# CSGO: CONTENT-STYLE COMPOSITION IN TEXT-TO-IMAGE GENERATION

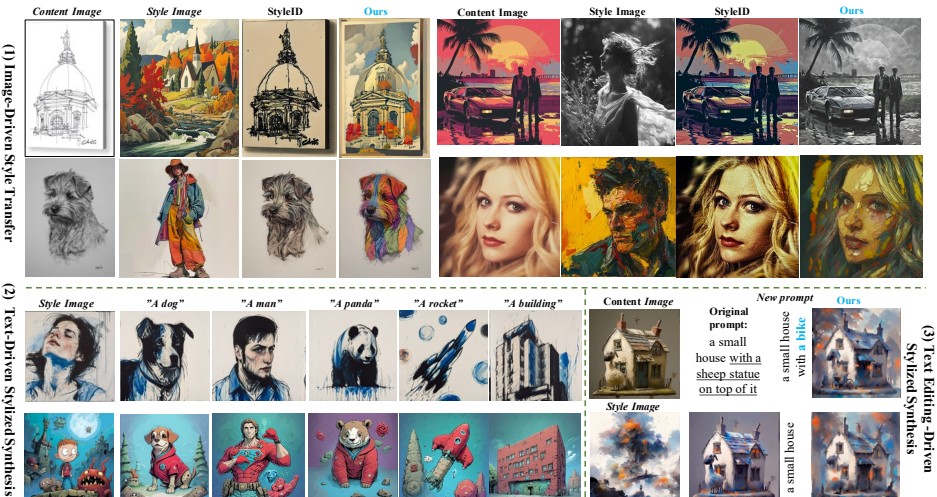

Figure 1: (1) Comparison of the style transfer results of the proposed method with the recent state-of-the-art method StyleID Chung et al. (2024). (2) Our CSGO achieves high-quality text-driven stylized synthesis. (3) Our CSGO achieves high-quality text editing-driven stylized synthesis, where content, where the italicized text (*e.g.*, *content image*) is the CSGO input.

## ABSTRACT

The diffusion model has shown exceptional capabilities in controlled image generation, which has further fueled interest in image style transfer. Existing works mainly focus on training free-based methods (e.g., image inversion) due to the scarcity of specific data. In this study, we present a data construction pipeline for content-style-stylized image triplets that generates and automatically cleanses stylized triplets. Based on this pipeline, we construct a dataset IMAGStyle, the first large-scale style transfer dataset containing 210k image triplets, available for the community to explore and research. Equipped with IMAGStyle, we propose a simple yet effective framework CSGO, a style transfer model based on end-to-end training, which explicitly decouples content and style features employing independent feature injection. Our CSGO implements image-driven style transfer, text-driven stylized synthesis, and text editing-driven stylized synthesis in the same model. We conduct extensive experiments on CSGO to validate the effectiveness of synthetic stylized data for style control. Meanwhile, ablation experiments show the effectiveness of CSGO.

## 1 INTRODUCTION

Recent advancements in diffusion models have significantly improved the field of text-to-image generation (Song et al., 2020; Ho et al., 2020). Models such as SD Rombach et al. (2022) excel at creating visually appealing images based on textual prompts, playing a crucial role in personalized content creation (Ruiz et al., 2023; Xu et al., 2024). Despite numerous studies on general controllability, image style transfer remains particularly challenging.

Image style transfer aims to generate a plausible target image by combining the content of one image with the style of another, ensuring that the target image maintains the original content's semantics while adopting the desired style (Jing et al., 2019; Deng et al., 2020). This process requires fine-grained control over content and style, involving abstract concepts like texture, color, composition, and visual quality, making it a complex and nuanced challenge (Chung et al., 2024).

A significant challenge in style transfer is the lack of a large-scale stylized dataset, which makes it impossible to train models end-to-end and results in suboptimal style transfer quality for non-end-to-end methods. Existing methods typically rely on training-free structures, such as DDIM inversion Song et al. (2020) or carefully tuned feature injection layers of pre-trained IP-Adapter (Ye et al., 2023). Methods like Plug-and-Play Tumanyan et al. (2023), VCT Cheng et al. (2023), and the state-of-the-art StyleID Chung et al. (2024) employ content image inversion and sometimes style image inversion to extract and inject image features into specifically designed layers. However, inverting content and style images significantly increases inference time, and DDIM inversion can lose critical information Mokady et al. (2023), leading to failures, as shown in Figure 1. InstantStyle Wang et al. (2024a) employs the pre-trained IP-Adapter. However, it struggles with accurate content control. Another class of methods relies on a small amount of data to train LoRA and implicitly decouple content and style LoRAs, such as ZipLoRA Shah et al. (2023) and B-LoRA Frenkel et al. (2024), which combine style and content LoRAs to achieve content retention and Style transfer. However, each image requires fine-tuning, and implicit decoupling reduces stability.

To overcome the above challenges, we start by constructing a style transfer-specific dataset and then design a simple yet effective framework to validate the beneficial effects of this large-scale dataset on style transfer. Initially, we propose a dataset construction pipeline for Content-Style-Stylized Image Triplets (CSSIT), incorporating both a data generation method and an automated cleaning process. Using this pipeline, we construct a large-scale stylized dataset, **IMAGStyle**, comprising 210K content-style-stylized image triplets. Next, we introduce an end-to-end trained style transfer framework, **CSGO**. Unlike previous implicit extractions, it explicitly uses independent content and style feature injection modules to achieve high-quality image style transformations. The framework simultaneously accepts style and content images as inputs and efficiently fuses content and style features using well-designed feature injection blocks. Benefiting from the decoupled training framework, once trained, CSGO realizes any form of arbitrary style transfer without fine-tuning at the inference stage, including sketch or nature image-driven style transfer, text-driven, text editing-driven stylized synthesis. Finally, we introduce a Content Alignment Score (CAS) to evaluate the quality of style transfer, effectively measuring the degree of content loss post-transfer. Extensive qualitative and quantitative studies validate that our proposed method achieves advanced zero-shot style transfer.

## 2    RELATED WORK

**Text-to-Image Model.** In recent years, diffusion models have garnered significant attention in the text-to-image generation community due to their powerful generative capabilities demonstrated by early works (Dhariwal & Nichol, 2021; Ramesh et al., 2022). Owing to large-scale training Schuhmann et al. (2022), improved architectures Radford et al. (2021); Peebles & Xie (2023), and latent space diffusion mechanisms, models like Stable Diffusion have achieved notable success in text-to-image generation (Ramesh et al., 2022). The focus on controllability in text-to-image models has grown in response to practical demands. Popular models such as ControlNet Zhang et al. (2023a), T2Iadapter Mou et al. (2024), and IP-Adapter Ye et al. (2023) introduce additional image conditions to enhance controllability. These models use sophisticated feature extraction methods and integrate these features into well-designed modules to achieve layout control. In this paper, we present a style transfer framework, CSGO, based on an image-conditional generation model that can perform zero-shot style transfer.

**Style Transfer.** Style transfer has garnered significant attention and research due to its practical applications in art creation (Gatys et al., 2016). Early methods, both optimization-based Gatys et al. (2016) and inference-based Chen et al. (2017); Dumoulin et al. (2016), are limited by speed constraints and the diversity of style transfer. The AdaIN approach Huang & Belongie (2017), which separates content and style features from deep features, has become a representative method for style transfer, inspiring a series of techniques using statistical mean and variance (Chen et al., 2021; Hertz et al., 2024). Additionally, transformer-based methods such as StyleFormer Wu et al. (2021) and

StyTR$^2$ Deng et al. (2022) improve content bias. However, these methods primarily focus on color or stroke transfer and face limitations in arbitrary style transfer.

Currently, inversion-based Style Transfer (InST) Zhang et al. (2023b) is proposed to obtain inversion latent of style image and manipulate attention maps to edit generated Images. However, DDIM (Denoising Diffusion Implicit Models) inversion results in content loss and increased inference time (Song et al., 2020). Hertz *et al.* explore self-attention layers using key and value matrices for style transfer (Hertz et al., 2024). DEADiff Qi et al. (2024) and StyleShot Junyao et al. (2024) are trained through a two-stage style control method. However, it is easy to lose detailed information within the control through sparse lines. InstantStyle Wang et al. (2024a;b) to achieve high-quality style control through pre-trained prompt adapter Ye et al. (2023) and carefully designed injection layers. However, these methods struggle with achieving high-precision style transfer and face limitations related to content preservation. Some fine-tuning approaches, such as LoRA Hu et al. (2021), DB-LoRA Ryu, Zip-LoRA Shah et al. (2023), and B-LoRA Frenkel et al. (2024), enable higher-quality style-controlled generation but require fine-tuning for different styles and face challenges in achieving style transfer. Our proposed method introduces a novel style transfer dataset and develops the CSGO framework, achieving high-quality arbitrary image style transfer without the need for fine-tuning.

## 3 DATA PIPELINE

In this section, we first introduce the proposed pipeline for constructing content-style-stylized image triplets. Then, we describe the constructed IMAGStyle dataset in detail.

### 3.1 PIPELINE FOR CONSTRUCTING CONTENT-STYLE-STYLIZED IMAGE TRIPLETS

The lack of a large-scale open-source dataset of content-style-stylized image pairs (CSSIT) in the community seriously hinders the research on style transfer. In this work, we propose a data construction pipeline that automatically constructs and cleans to obtain high-quality content-style-stylized image triplets, given only arbitrary content images and style images. The pipeline contains two steps: (1) stylized image generation and (2) stylized image cleaning.

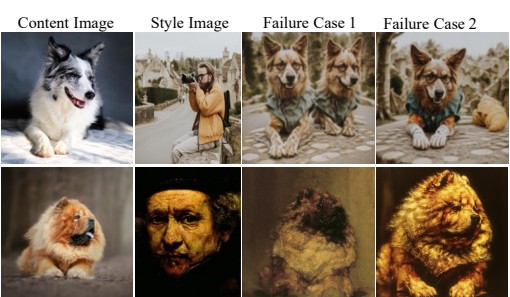

Figure 2: Failure cases in step (1), which fail to maintain the layout of the content image.

**Stylized image generation.** Given an arbitrary content image $C$ and an arbitrary style image $S$, the goal is to generate a stylized image $T$ that preserves the content of $C$ while adopting the style of $S$. We are inspired by B-LoRA Frenkel et al. (2024), which finds that content LoRA and style LoRA can be implicitly separated by SD-trained LoRA, preserving the original image's content and style information, respectively. Therefore, we first train a large number of Lo-RAs with lots of content and style images. To ensure that the content of the generated image $T$ is aligned to $C$ as much as possible, the loRA for $C$ is trained using only one content image $C$. Then, Each trained loRA is decomposed into a content LoRA and a style LoRA through implicit separate mentioned by work Frenkel et al. (2024). Finally, the content LoRA of image $C$ is

---

**Algorithm 1** Pipeline of Constructing CSSIT

**Input:** content images $Set_{content}$, style images $Set_{style}$
**Output:** Content-style-stylized image triplets $Set$
1: **for** each $C \in Set_{content}$ **do**
2:     $C_{LoRA} \leftarrow$ Train LoRA for $C$
3:     $C_{LoRA}^{content} \leftarrow$ Separate content LoRA in $C_{LoRA}$
4:     **for** each $S \in Set_{style}$ **do**
5:         $S_{LoRA} \leftarrow$ Train LoRA for $S$
6:         $S_{LoRA}^{style} \leftarrow$ Separate style LoRA in $S_{LoRA}$
7:         $CS_{LoRA} \leftarrow$ Combine $C_{LoRA}^{content}$ and $S_{LoRA}^{style}$
8:         $T = \{T_1, T_2, ..., T_n\} \leftarrow$ Generate $n$ images by SDXL and $CS_{LoRA}$
9:         $CAS_1, CAS_2, ..., CAS_n \leftarrow$ Compute CAS for each generated image based on Equ.( 1)
10:         $i \leftarrow$ Obtain the index of the minimum value of all CAS
11:         $Set$.append([$C, S, T_i$])
12:     **end for**
13: **end for**
14: **return** $Set$

---

combined with the style LoRA of $S$ to generate the target images $T = \{T_1, T_2, ..., T_n\}$ using the base model. However, the implicit separate approach is unstable, resulting in the content and style LoRA not reliably retaining content or style information. This manifests itself in the form of the generated

image $T_i$, which does not always agree with the content of $C$, as shown in Figure 2. Therefore, it is necessary to filter $T$, sampling the most reasonable $T_i$ as the target image.

**Stylized image cleaning.** Slow methods of cleaning data with human involvement are unacceptable for the construction of large-scale stylized data triplets. To this end, we develop an automatic cleaning method to obtain the ideal and high-quality stylized image $T$ efficiently. First, we propose a content alignment score (CAS) that effectively measures the content alignment of the generated image with the content image. It is defined as the feature distance between the content semantic features (without style information) of the generated image and the original content image. It is represented as follows:

$$CAS_i = \|Ada(\phi(C)) - Ada(\phi(T_i))\|^2,  \quad (1)$$

where $CAS_i$ denotes the content alignment score of generated image $T_i$, $\phi(\cdot)$ denotes image encoder. We compare the mainstream feature extractors and the closest to human filtering results is DINO-V2 (Li et al., 2023). $Ada(F)$ represents a function of feature $F$ to remove style information. We follow AdaIN Huang & Belongie (2017) to express style information by mean and variance. It is represented as follows:

$$Ada(F) = \frac{F - \mu(F)}{\rho(F)},  \quad (2)$$

where $\mu(F)$ and $\rho(F)$ represent the mean and variance of feature $F$. Obviously, a smaller CAS indicates that the generated image is closer to the content of the original image. In Algorithm 1, we provide a pseudo-code of our pipeline.

### 3.2 IMAGStyle Dataset Details

**Content Images.** To ensure that the content images have clear semantic information and facilitate separating after training, we employ the saliency detection datasets, MSRA10K Cheng et al. (2015; 2013) and MSRA-B Jiang et al. (2013), as the content images. In addition, for sketch stylized, we sample 1000 sketch images from ImageNet-Sketch Wang et al. (2019) as content images. The category distribution of content images is shown in Figure 3.

We use BLIP Li et al. (2023) to generate a caption for each content image. A total of 11,000 content images are trained and used as content LoRA.

**Style Images.** To ensure the richness of the style diversity, we sample 5000 images of different painting styles (history painting, portrait, genre painting, landscape, and still life) from the Wikiart dataset (Saleh & Elgammal, 2016). In addition, we generated 5000 images using Midjourney covering diverse styles, including Classical, Modern, Romantic, Realistic, Surreal, Abstract, Futuristic, Bright, Dark styles etc. A total of 10,000 style images are used to train style LoRA.

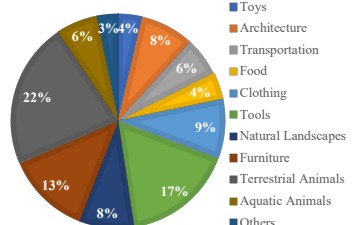

Figure 3: Distribution of content images.

**Dataset.** Based on the pipeline described in Section 3.1, we construct a style transfer dataset, **IMAGStyle**, which contains 210K content-style-stylized image triplets as training dataset. Furthermore, we collect 248 content images from the web containing images of real scenes, sketched scenes, faces, and style scenes, as well as 206 style images of different scenes as testing dataset. For testing, each content image is transferred to 206 styles. This dataset will be used for community research on style transfer and stylized synthesis.

## 4 Approach

### 4.1 CSGO framework

The proposed style transfer model, CSGO, shown in Figure 4, aims to achieve arbitrary stylization of any image without fine-tuning, including sketch and natural image-driven style transfer, text-driven stylized synthesis, and text editing-driven stylized synthesis. Benefiting from the proposed IMAGStyle dataset, the proposed CSGO supports an end-to-end style transfer training paradigm. To ensure effective style transfer and accurate content preservation, we carefully design the content and style control modules. In addition, to reduce the risk that the content image leaks style information or

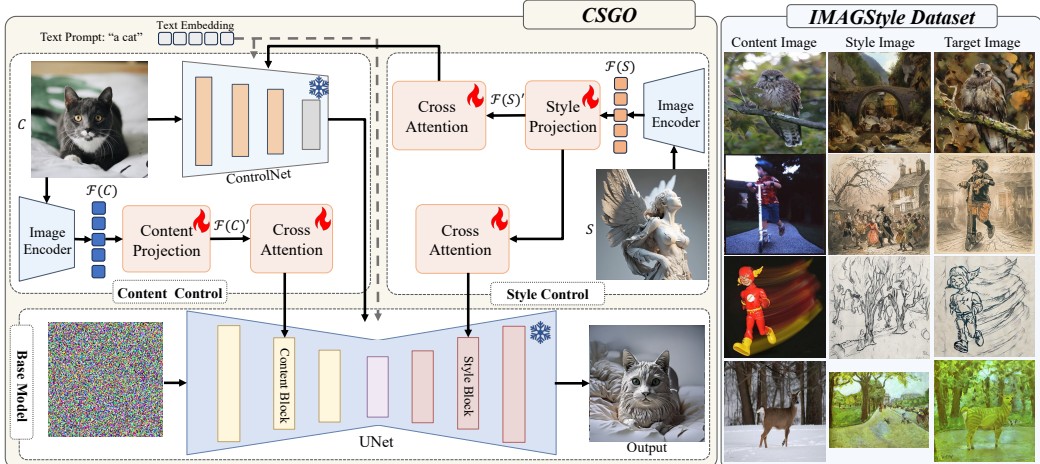

Figure 4: (a) Left: Overview of the proposed end-to-end style transfer framework CSGO. (b) Right: Samples from our IMAGStyle dataset. For content control, firstly, we use an image encoder and projection layer to extract the content features $\mathcal{F}(C)'$, and inject them into the down-sampling blocks through an additional cross attention layer. Secondly, we use the pre-trained ControlNet to extract the fine-grained information of the content image and inject it into the up-sampling blocks to ensure the content preservation. For style control, we use an image encoder to extract the style features $\mathcal{F}(S)'$ of the style image $S$, which are injected into the up-sampling blocks of the ControlNet model and the base model using the cross attention layer, respectively. Finally, in the inference stage, the styled results can be generated efficiently without fine-tuning.

the style image leaks content, the content control and style control modules are explicitly decoupled, and the corresponding features are extracted independently. To be more specific, we categorize our CSGO into two main components and describe them in detail.

**Content Control.** The purpose of content control is to ensure that the stylized image retains the semantics, layout, and other features of the content image. To this end, we carefully designed two ways of content control. First, we implement content control through pre-trained ControlNet Zhang et al. (2023a), whose input is the content image and the corresponding caption. We leverage the capabilities of the specific content-controllable model(Tile ControlNet) to reduce the data requirements and computational costs of training content retention from scratch Following the ControlNet, the output of ControlNet is directly injected into the up-sampling blocks of the base model (pre-trained UNet in SD) to obtain fusion output $D_i' = D_i + \delta_c \times C_i$, $D_i$ denotes the output of $i$-th block in the base model, $C_i$ denotes the output of $i$-th block in ControlNet, $\delta_c$ represents the fusion weight.

In addition, to achieve content control in the down-sampling blocks of the base model, we utilize an additional learnable cross-attention layer to inject content features into down blocks. Specifically, we use pre-trained CLIP image encoder Radford et al. (2021) and a learnable projection layer to extract the semantic feature $\mathcal{F}(C)'$ of the content image. Then, we utilize an additional cross-attention layer to inject the extracted content features into the down-sampling blocks of the base model, *i.e.*, $D_C' = D + \lambda_c \times D_C$, $D$ denotes the output of in the base model, $D_C$ denotes the output of content IP-Adapter, $\lambda_c$ represents the fusion weight (Ye et al., 2023). These two content control strategies ensure small content loss during the style transfer.

**Style Control.** To ensure that the proposed CSGO has strong style control capability, we also design two simple yet effective style control methods. Generally, we feed the style images into a pre-trained image encoder to extract the original embedding $\mathcal{F}(S) \in \mathbb{R}^{o \times d}$ and map them to the new embedding $\mathcal{F}(S)' \in \mathbb{R}^{t \times d}$ through the Style Projection layer. Here, $o$ and $t$ represent the token number of original and new embeddings, $d$ denotes the dimension of $\mathcal{F}(S)$. For style projection, we employ the Perceiver Resampler structure Alayrac et al. (2022) to obtain more detailed style features. Then, we utilize an additional cross-attention layer to inject the new embedding into the up-sampling blocks of the base model.

Furthermore, we note that relying only on the injection of the up-sampling blocks of the base model weakens the style control since ControlNet injections in the content control may leak style information

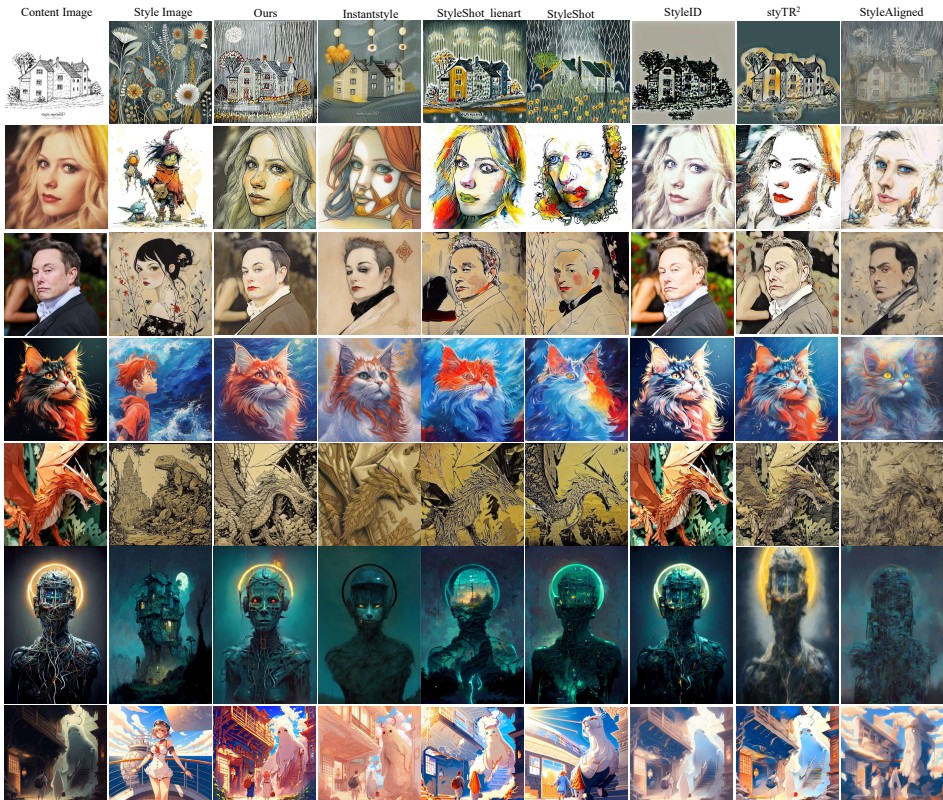

Figure 5: Comparison of image-driven style transfer results. Zoomed in for the best viewing.

of the content image $C$. For this reason, we propose to use an independent cross attention module to simultaneously inject style features into, and the fusion weight is $\lambda_s$, as shown in Figure 4. The insight of this is to pre-adjust the style of the content image using style features making the output of the Controlnet model retain the content while containing the desired style features.

In summary, the proposed CSGO framework explicitly learns separate feature processing modules that inject style and content features into different locations of the base model, respectively. Despite its simplicity, CSGO achieves state-of-the-art style transfer results.

## 4.2 MODEL TRAINING AND INFERENCE.

**Training.** Based on the proposed dataset, IMAGStyle, our CSGO is the first implementation of end-to-end style transfer training. Given a content image $C$, a caption $P$ of the content image, a style image $S$, and a target image $T$, we train a style transfer network based on a pre-trained diffusion model. Our training objective is to model the relationship between the styled image $T$ and Gaussian noise under content and style image conditions, which is represented as follows:

$$\mathcal{L} = \mathbb{E}_{z_0,t,P,C,S,\epsilon\sim\mathcal{N}(0,1)} \left[ \|\epsilon - \epsilon_\theta \left(z_t, t, C, S, P\right)\|^2 \right], \tag{3}$$

where $\varepsilon$ denotes the random sampled Gaussian noise, $\varepsilon_\theta$ denotes the trainable parameters of CSGO, $t$ represents the timestep. Note that the latent latent $z_t$ is constructed with a style image $T$ during training, $z_t = \sqrt{\bar{\alpha}_t}\psi(T) + \sqrt{1 - \bar{\alpha}_t}\varepsilon$, where $\psi(\cdot)$ mapping the original input to the latent space function, $\bar{\alpha}_t$ is consistent with diffusion models (Song et al., 2020; Ho et al., 2020). We randomly drop content image and style image conditions in the training phase to enable classifier-free guidance in the inference stage.

**Inference.** During the inference phase, we employ classifier-free guidance. The output of timestep $t$ is indicated as follows:

$$\hat{\epsilon}_\theta(z_t, t, C, S, P) = w\epsilon_\theta(z_t, t, C, S, P) + (1 - w)\epsilon_\theta(z_t, t), \tag{4}$$

where $w$ represents the classifier-free guidance factor (CFG).

Table 1: Comparison of style similarity (CSD) and content alignment (CAS) with recent state-of-the-art methods on the test dataset.

| | StyTR$^2$ (Deng et al., 2022) | Style-Aligned (Hertz et al., 2024) | StyleID (Chung et al., 2024) | InstantStyle (Wang et al., 2024a) | StyleShot (Junyao et al., 2024) | StyleShot-lineart (Junyao et al., 2024) | CSGO Ours |
|---|---|---|---|---|---|---|---|
| CSD (↑) | 0.2695 | 0.4274 | 0.0992 | 0.3175 | 0.4522 | 0.3903 | 0.5146 |
| CAS (↓) | 0.9699 | 1.3930 | 0.4873 | 1.3147 | 1.5105 | 1.0750 | 0.8386 |

# 5 EXPERIMENTS

## 5.1 EXPERIMENTAL SETUP

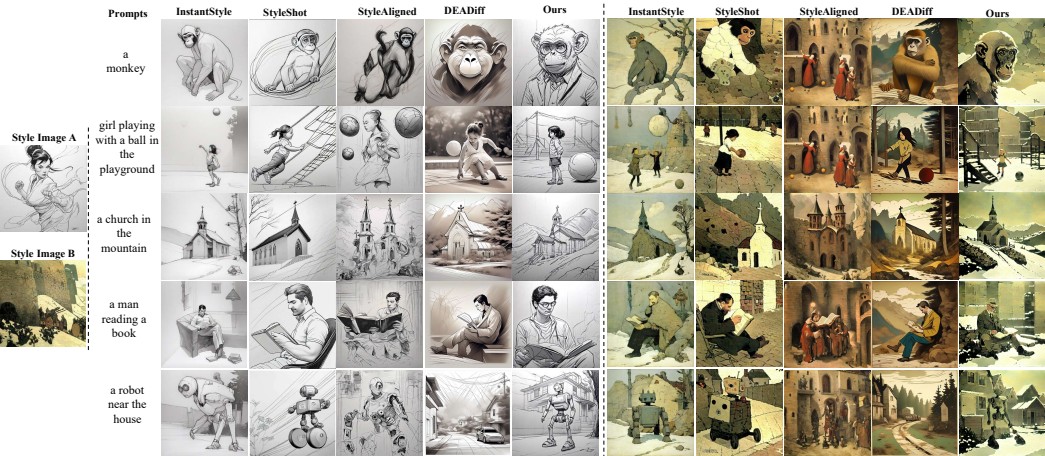

Figure 6: Comparison of generation results for text-driven stylized synthesis with recent methods.

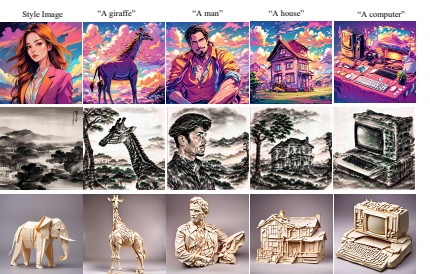

Figure 7: Generated results of the proposed CSGO in text-driven stylized synthesis.

Figure 8: The generated results of the proposed CSGO in text editing-driven stylized synthesis.

**Setup.** For the IMAGstyle dataset, during the training phase, we suggest using 'a [vcp]' as a prompt for content images and 'a [stp]' as a prompt for style images. The rank is set to 64 and each B-loRA is trained with 1000 steps. During the generation phase, we suggest using 'a [vcp] in [stv] style' as the prompt. For the CSGO framework, we employ *stabilityai/stable-diffusion-xl-base-1.0* as the base model, pre-trained *ViT-H* as image encoder, and *TTPlanet/TTPLanet_SDXL_Controlnet_Tile_Realistic* as ControlNet. we uniformly set the images to $512 \times 512$ resolution. The drop rate of text, content image, and style image is 0.15. The learning rate is 1e-4. During training stage, $\lambda_c = \lambda_s = \delta_c = 1.0$. During inference stage, we suggest $\lambda_c = \lambda_s = 1.0$ and $\delta_c = 0.5$. Our experiments are conducted on 8 NVIDIA H800 GPUs (80GB) with a batch size of 20 per GPU and trained 80000 steps.

**Datasets and Evaluation.** We use the proposed IMAGStyle as a training dataset and use its testing dataset as an evaluation dataset. It is worth noting that the style transfer task, unlike the rest of the style control tasks, requires a trade-off between content retention and style quality at the same time. We use the CSD score Somepalli et al. (2024) as an evaluation metric to evaluate the style similarity. Meanwhile, we employ the proposed content alignment score (CAS) as an evaluation metric to evaluate the content similarity.

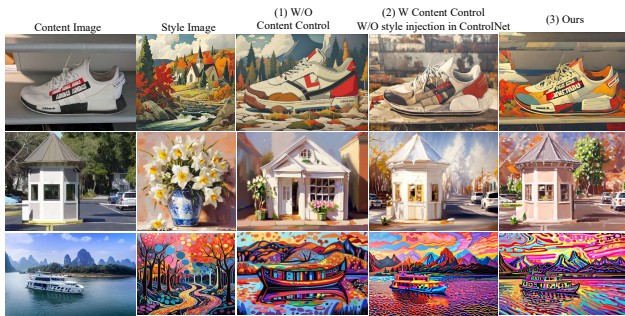

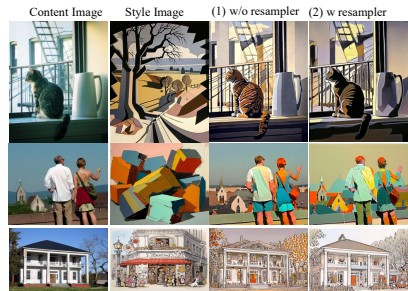

Figure 9: Ablation studies of content control and style control.

Figure 10: Ablation studies of style image projection.

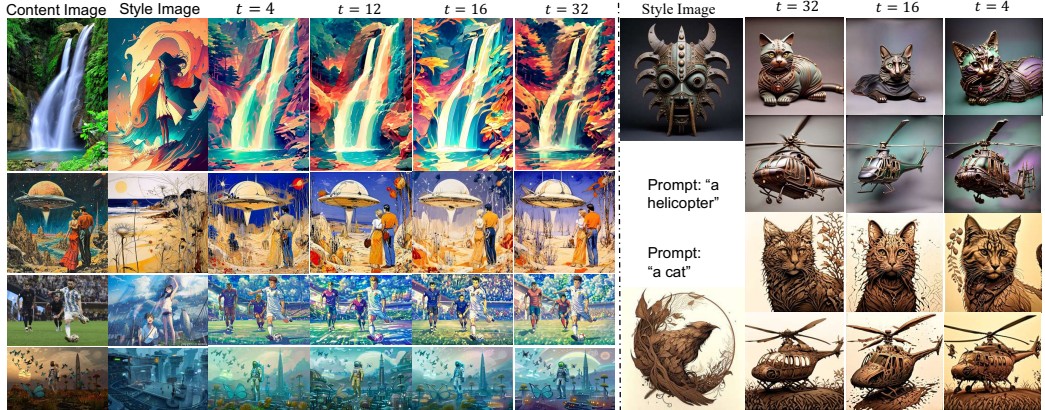

Figure 11: Ablation studies of style token number $t$. Left: Image style transfer results. Right: Text-driven stylized synthesis results.

**Baselines.** We compare recent advanced inversion-based StyleID Chung et al. (2024), StyleAligned Hertz et al. (2024) methods, and StyTR$^2$ Deng et al. (2022) based on the Transformer structure. In addition, we compare Instantstyle Wang et al. (2024a) and StyleShot (and their fine-grained control method StyleShot-lineart) Junyao et al. (2024) that introduce ControlNet and IPAdapter structures as baselines. For text-driven style control task, we also introduce DEADiff Qi et al. (2024) as a baseline.

## 5.2 EXPERIMENTAL RESULTS

**Image-Driven Style Transfer.** In Table 1, we demonstrate the CSD scores and CAS of the proposed method with recent advanced methods for the image-driven style transfer task. In terms of style control, our CSGO achieves the highest CSD score, demonstrating that CSGO achieves state-of-the-art style control. Due to the decoupled style injection approach, the proposed CSGO effectively extracts style features and fuses them with high-quality content features. As illustrated in Figure 5, Our CSGO precisely transfers styles while maintaining the semantics of the content in natural, sketch, face, and art scenes. More results for style control can be found in the supplementary material.

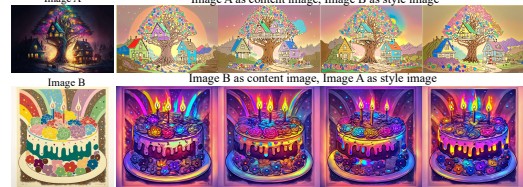

Figure 12: Results of content-style cycle transfer.

In terms of content retention, it can be observed that StyleID Chung et al. (2024) and StyleAligned Hertz et al. (2024), which are based on inversion, maintain the original content too strongly in sketch style transfer scenarios (CAS is very low). However. they are unable to inject style information since CSD score is low. InstantStyle Wang et al. (2024a) and StyleShot Junyao et al. (2024) (including Lineart), which use lines to control the content, are affected by the level of detail of the lines and have different degrees of loss of content (such as face scenes). The proposed CSGO directly utilizes all the information of the content image, and content preservation is optimal.

The quantitative results in Table 1 also show that the proposed CSGO maintains high-quality content retention with precise style transfer. It is worth noting that it is possible to implement a content-style cycle transfer in CSGO (as shown in Figure 12; see the Appendix for more results).

**Text-Driven Stylized Synthesis.** The proposed method enables text-driven style control, *i.e.*, given a text prompt and a style image, generating images with similar styles. Figure 6 shows the comparison of the generation results of the proposed CSGO with the state-of-the-art methods. In a simple scene, it is intuitive to observe that our CSGO obeys textual prompts more. The reason for this is that thanks to the explicit decoupling of content and style features, style images only inject style information without exposing content. In addition, in complex scenes, thanks to the well-designed style feature injection block, CSGO enables optimal style control while converting the meaning of text. As illustrated in Figure 7, we demonstrated more results.

**Text editing-Driven Stylized Synthesis.** The proposed CSGO supports text editing-driven style control. As shown in Figure 8, in the style transfer, we maintain the semantics and layout of the original content images while allowing simple editing of the textual prompts. The above excellent results demonstrate that the proposed CSGO is a powerful framework for style control.

## 5.3 ABLATION STUDIES.

**Content control and style control.** We discuss the impact of the two feature injection methods, as shown in Figure 9. The content image must be injected via ControlNet injection to maintain the layout while preserving the semantic information. If content images are injected into the base model only through an additional Cross attention layer, only semantic information is guaranteed, while the

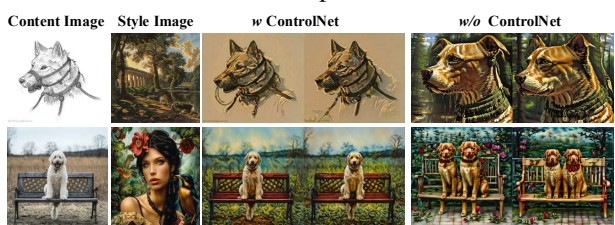

Figure 13: Ablation studies of ControlNet.

full content information is not preserved (Figure 9(1)). After introducing the ControlNet injection, the quality of content retention improved, as shown in Figure 13. However, if the style features are injected into base UNet only without ControlNet injection, this weakens the style of the generated images, which can be observed in the comparison of Figure 9(2) and (3). Therefore, the proposed CSGO pre-injects style features in the ControlNet branch to further fuse the style features to enhance the transfer effect.

**Style image projection layer.** The style image projection layer can effectively extract style features from the original embedding. We explore the normal linear layer and the Resampler structure, and the experimental results are shown in Figure 10. Using the Resampler structure captures more detailed style features while avoiding content leakage.

**Token number.** We explore the effect of the number of token $t$ in the style projection layer on the results of style transfer and text-driven style synthesis. The experimental results are shown in Figure 10, where the style control becomes progressively better as $t$ increases. This is in line with our expectation that $t$ influences the quality of feature extraction. A larger $t$ means that the projection layer can extract richer style features.

**The impact of content scale $\delta_c$.** As shown in Figure 14, when $\delta_c$ is small, the content feature injection is weak, and CSGO obeys the textual prompts and style more. As $\delta_c$ increases, the quality of content retention becomes superior. However, we notice that when $\delta_c$ is large (e.g., 0.9 and 1.2), the style information is severely weakened.

**The impact of CFG scale.** Classifier-free guidance enhances the capabilities of the text-to-image model. The proposed CSGO is similarly affected by the strength of CFG scale. As shown in Figure 14, the introduction of CFG enhances the style transfer effect.

**The impact of style scale $\lambda_s$ and content scale $\lambda_c$.** The style scale affects the degree of style injection. Figure 14 shows that if the style scale is less than 1.0, the style of the generated image is severely weakened. We suggest that the style scale should be between 1.0 and 1.5. Content control in the down-sampling blocks utilizes the semantic information of the content image to reinforce the accurate retention of content. Figure 14 shows that $\lambda_c$ is most effective when it is near 1.0.

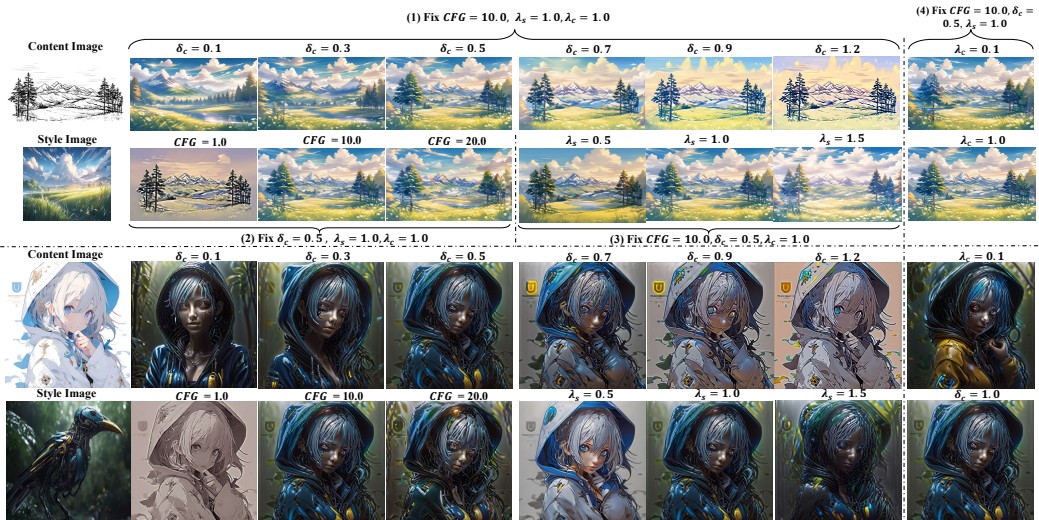

Figure 14: Ablation studies of content scale $\delta_c$, CFG, content scale $\lambda_c$, and style scale $\lambda_s$.

In style transfer, the retention of content and style varies from person to person. We can set the hyperparameter content scale $\delta_c$ so that the generated result meets the expectation. As shown in Figure 15, different levels of detailed information can be retained by different content scales to meet different design requirements.

# 6 CONCLUSION.

We first propose a pipeline for the construction of content-style-stylized image triplets. Based on this pipeline, we construct the first large-scale style transfer dataset, IMAGStyle, which contains 210K image triplets and covers a wide range of style scenarios. To validate the impact of IMAGStyle on style transfer, we propose CSGO, a simple but highly effective end-to-end training style transfer framework, and we verify that the proposed CSGO can simultaneously perform image

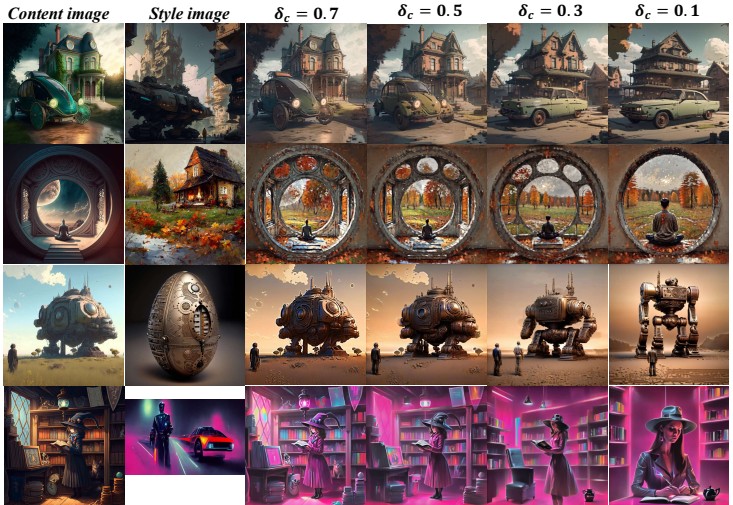

Figure 15: Effects of content scale in style transfer task.

style transfer, text-driven style synthesis, and text editing-driven style synthesis tasks in a unified framework. Extensive experiments validate the beneficial effects of IMAGStyle and CSGO for style transfer. We hope that our work will inspire the research community to further explore stylized research.

**Future work.** Although the proposed dataset and framework achieve very advanced performance, there is still room for improvement. Due to time and computational resource constraints, we constructed only 210K data triplets. We believe that by expanding the size of the dataset, the style transfer quality of CSGO will be even better. Meanwhile, the proposed CSGO framework is a basic version, which only verifies the beneficial effects of generative stylized datasets on style transfer. We believe that the quality of style transfer can be further improved by optimizing the style and content feature extraction and fusion methods.

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
