# Appendix of CSGO: Content-Style Composition in Text-to-Image Generation

## 1 Details.

### 1.1 Preliminaries

**Stable Diffusion.** The backbone of the stable diffusion model Rombach et al. (2022) uses CLIP Radford et al. (2021) as a text encoder and the UNet structure Ronneberger et al. (2015) as a latent denoising network. We refer to the pre-trained UNet as the base model. In general, the U-Net contains multiple down-sampling blocks, a middle block, and multiple up-sampling blocks (Podell et al., 2023). Studies on the controllability of diffusion models usually inject control features into the base model.

**ControlNet.** ControlNet models have been developed for image conditions such as depth map, canny, and sketch, which effectively enhance the controllability of text-to-image models. ControlNet Zhang et al. (2023) takes images as the control condition, trains zero-convolution layers and replicates the encoder (down-sampling blocks and middle block) of the base model, and injects the resulting outputs of each block into the up-sampling blocks and middle block correspondingly. Controlnet output features are directly weighted with base model features.

**IP-Adapter.** IP-Adapter Ye et al. (2023) implements image prompt features injected into the text-to-image model by decoupling the cross attention module. In general, image prompts are first obtained as image embeddings by a pre-trained encoder, and then mapped to the Key matrix and Value matrix of the attention. Then, they interact with the Query matrix of the base model's attention layer and weight the outputs with the original outputs. IP-Adapter's simplicity and effectiveness in injecting image conditions have received wide attention from the community.

### 1.2 Datasets.

**CAS metric.** In Figure 1, we show a case where some images are generated by a combination of content LoRA and style LoRA, and the CAS metrics of each generated image are computed. The image identified in bold (second row, third column) has the lowest score and serves as a candidate for the target image. From Figure 1, it can be cleanly seen that CAS can effectively filter the generated images that do not match the content image in terms of pose, size, and shape.

In addition, we show 10 sets of CAS filtering examples in Figures 18, 19, 20, 21, 22. We show the original content image, the style image, and the best result obtained by CAS filtering (target image), and a large number of original results generated by B-LoRA. These cases show that CAS can clean illogical generated graphs for pose, size, and so on. However, we emphasize that since B-LoRA is actually more stable for the generation of styles, it is up to us to filter the images with CSD. In our experiments, it is possible to filter using only CAS without CSD.

In order to construct IMAGStyle, we trained a total of 11,000 content LoRAs and 10,000 style LoRAs. Theoretically, the combination of these LoRAs can generate $11,000 \times 10,000 = 110M$ images. However, image generation is very time and resource consuming. Secondly, not any combination of content LoRAs and style LoRAs can get the desired results. Therefore, we generated only 210k data triples in limited time. In the future, we expect to improve the quality of IMAGStyle further.

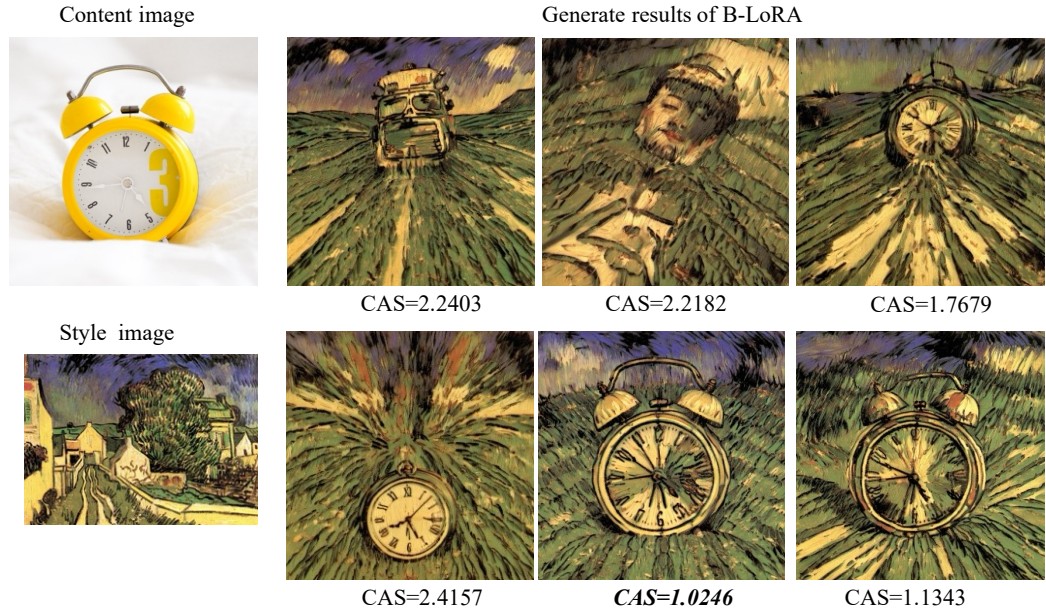

Figure 1: Example of generating a result that is filtered by CAS.

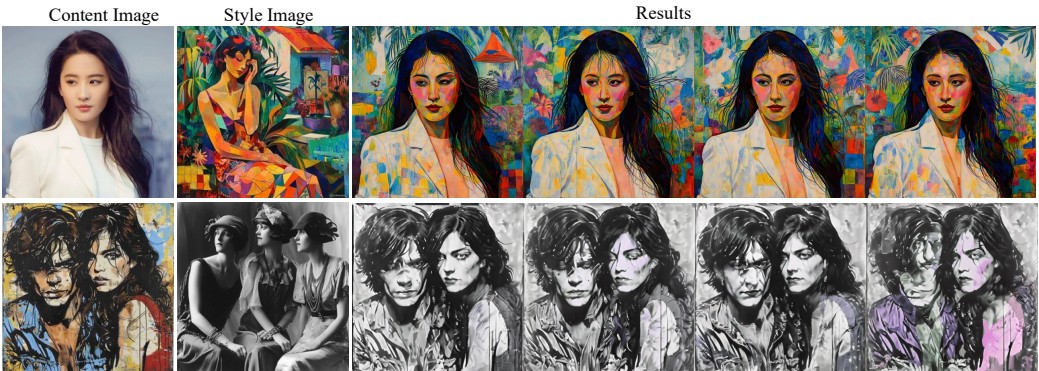

Figure 2: Failure cases.

## 2 MORE RESULTS.

### 2.1 FAILURE CASES.

As shown in Figure2, first, for real portrait stylization, as shown in the first row, there is a potential loss of facial identity. Portrait images can be difficult to collect due to the privacy issues involved, leading to some limitations in CSGO's style migration for real portraits. Second, despite incorporating styles into the ControlNet and base model, CSGO may still leak information, such as the original image's color. In the future, we aim to enhance the CSGO framework in several ways. First, we plan to use CSGO in conjunction with LoRA to improve the portrait segment of the IMAGStyle dataset and enhance portrait stylization capabilities. Second, we will redesign and train the content extractor and style encoder to minimize content leakage. Third, we intend to refine the current IPA fusion method and explore more effective approaches to style and content fusion. However, we acknowledge that these improvements may not be achievable in the short term.

We show the results of additional complementary experiments.

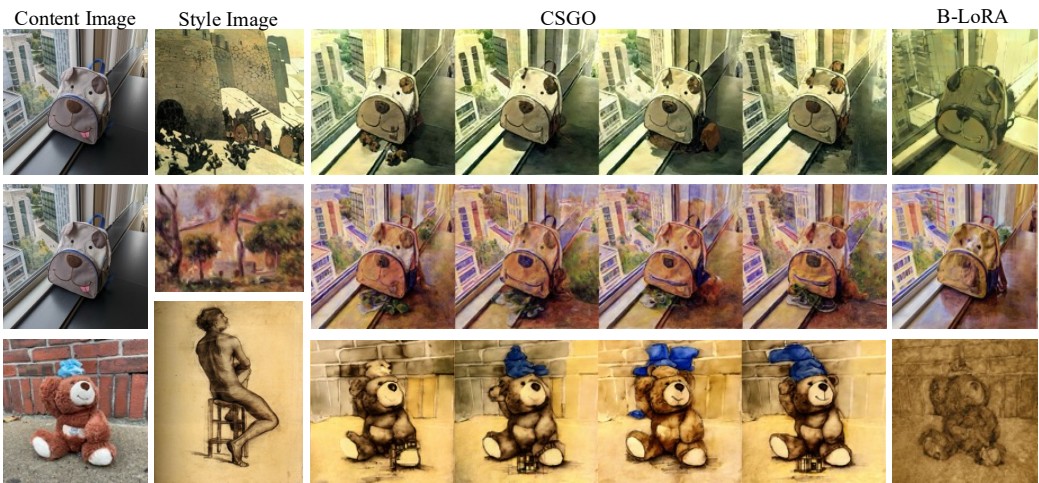

Figure 3: Comparison with LoRA.

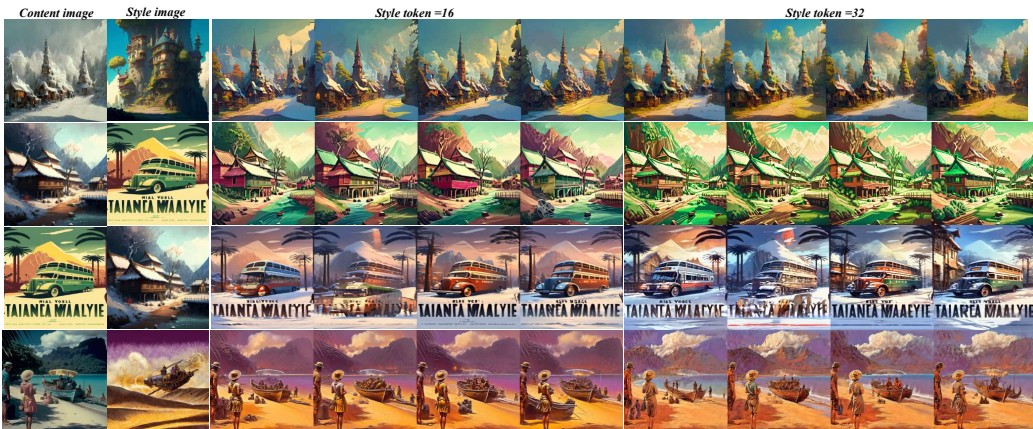

Figure 4: Ablation studies on style token.

- **Comparison with LoRA.** LoRA-based style transfer schemes need to be fine-tuned for different style images. We compared this with the current SOTA style transfer LORA scheme B-LoRA. We trained 3 sets of B-LoRA and the results are shown in Fig. 3. It can be seen that CSGO, which does not require re-fine-tuning, outperforms the style transfer results of B-LoRA.

- **Ablation studies on style token.** As expected, the number of style tokens influences the quality of style features. As shown in Figure 4, as the style token increases, the style transfer quality is better.

- **Line control and original image control.** We show in Figure 5 the results of controlling the content using lines and the original image, respectively. It can be clearly noticed that using lines loses detail information.

- Figures 6, 7, and 8 show sketch-driven style transfer results.

- Figures 9, 10, 11, and 12 shows image-driven style transfer results.

- Figure 13 shows content-style cycle transfer results.

- Figure 14 shows face image style transfer results.

- Figure 15 and Figure 16 show text-driven stylized synthesis results.

- Figure 17 shows text editing-driven stylized synthesis results.

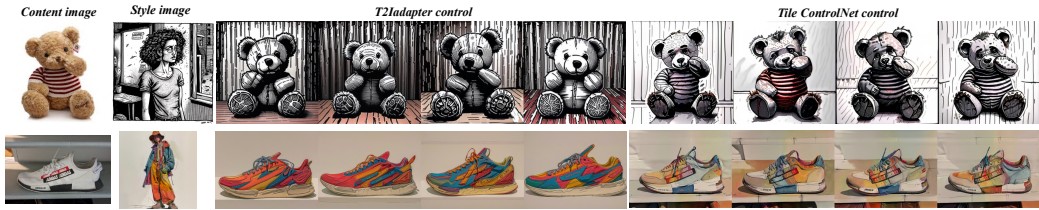

Figure 5: Results comparison between using line control content and original image control content.

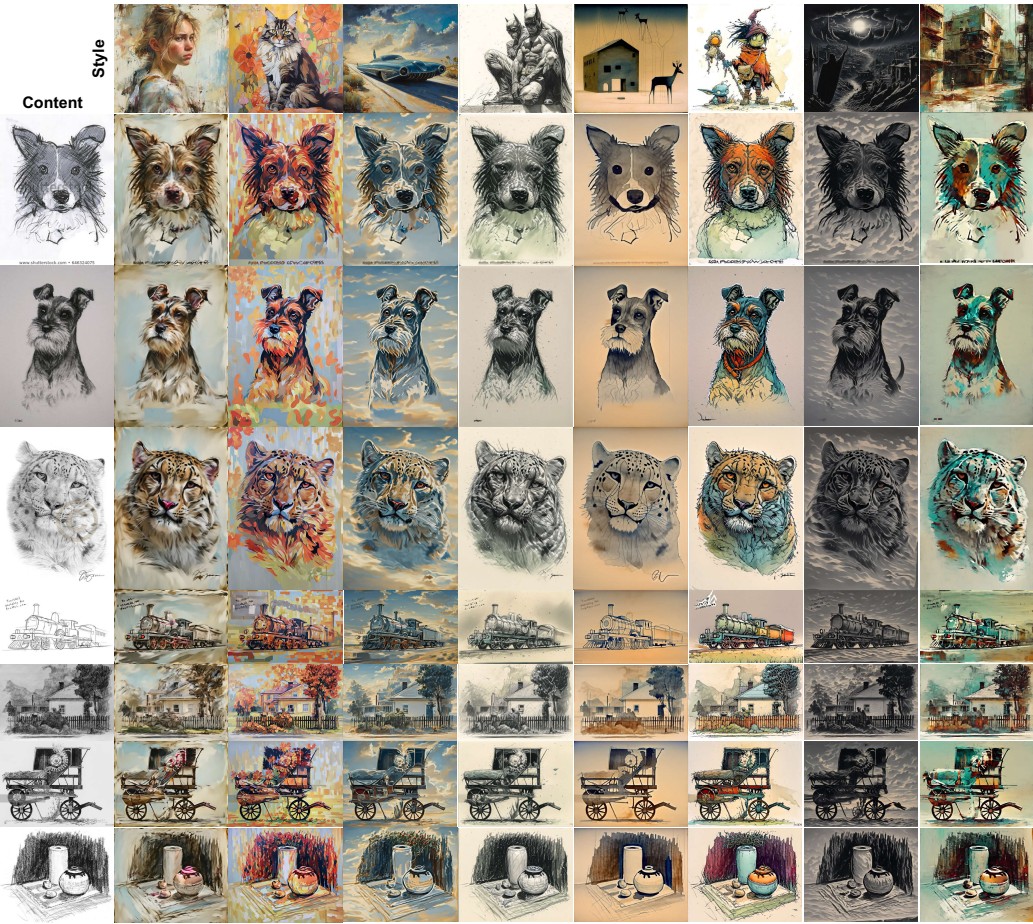

Figure 6: Sketch-driven style transfer results.

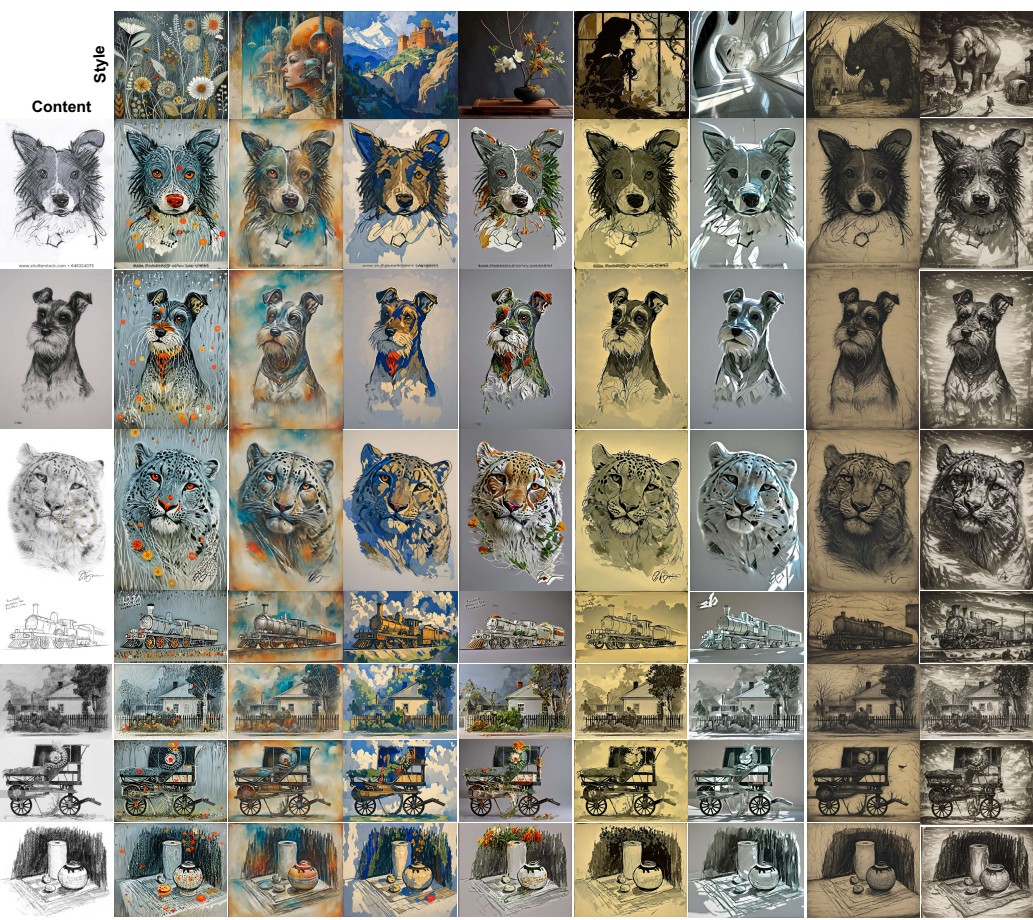

Figure 7: Sketch-driven style transfer results.

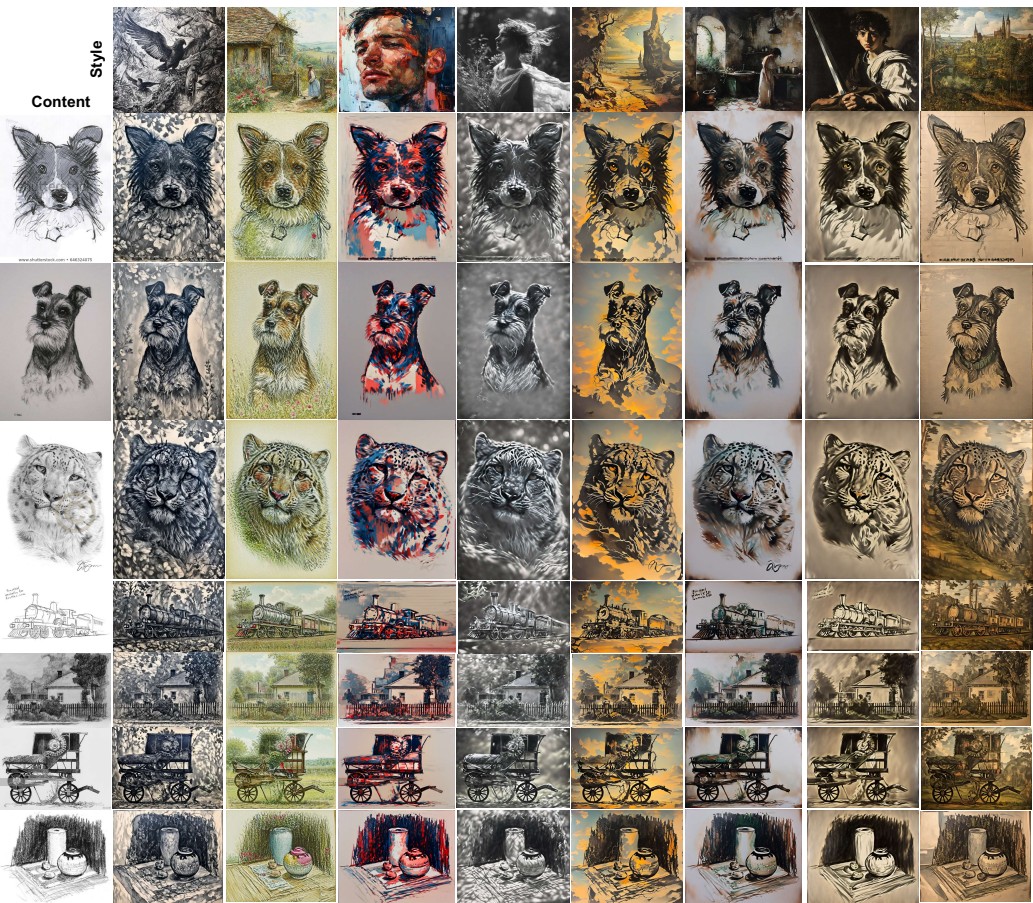

Figure 8: Sketch-driven style transfer results.

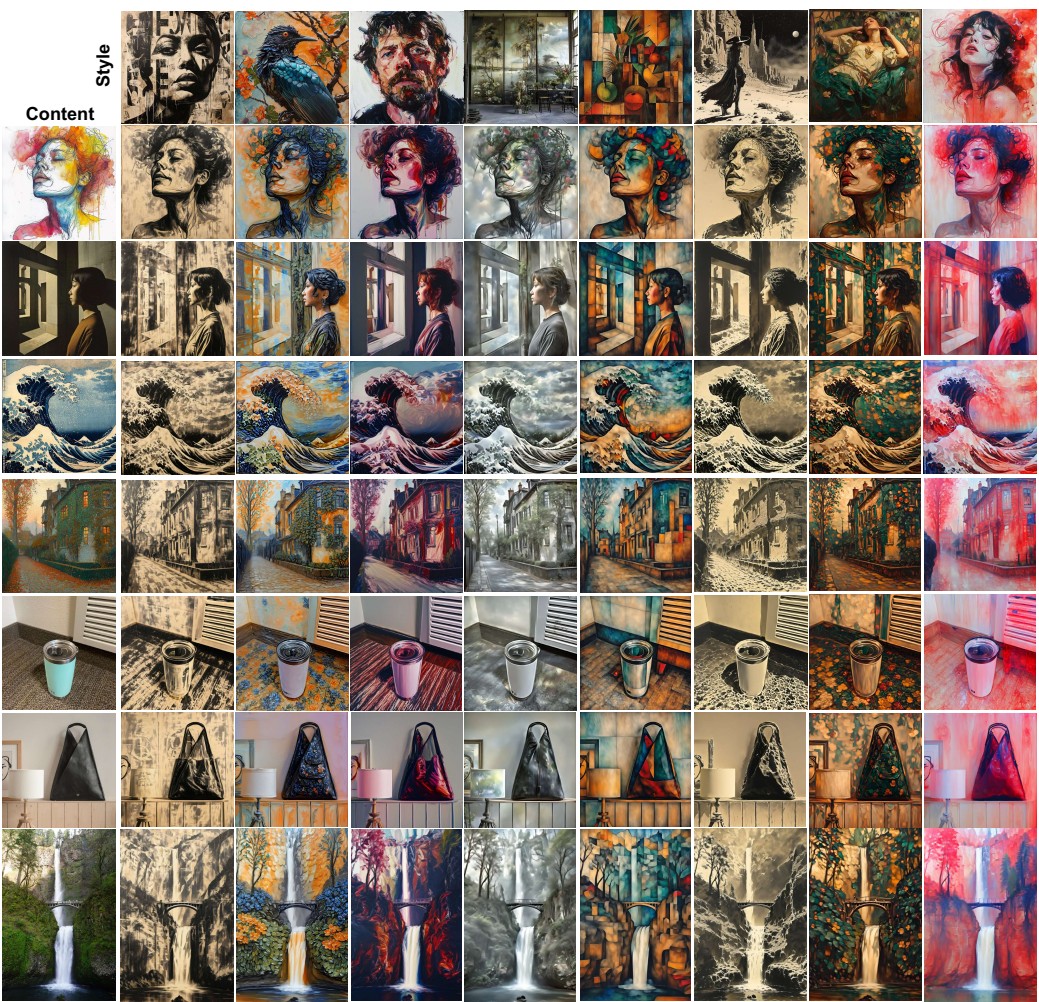

Figure 9: Image-driven style transfer results.

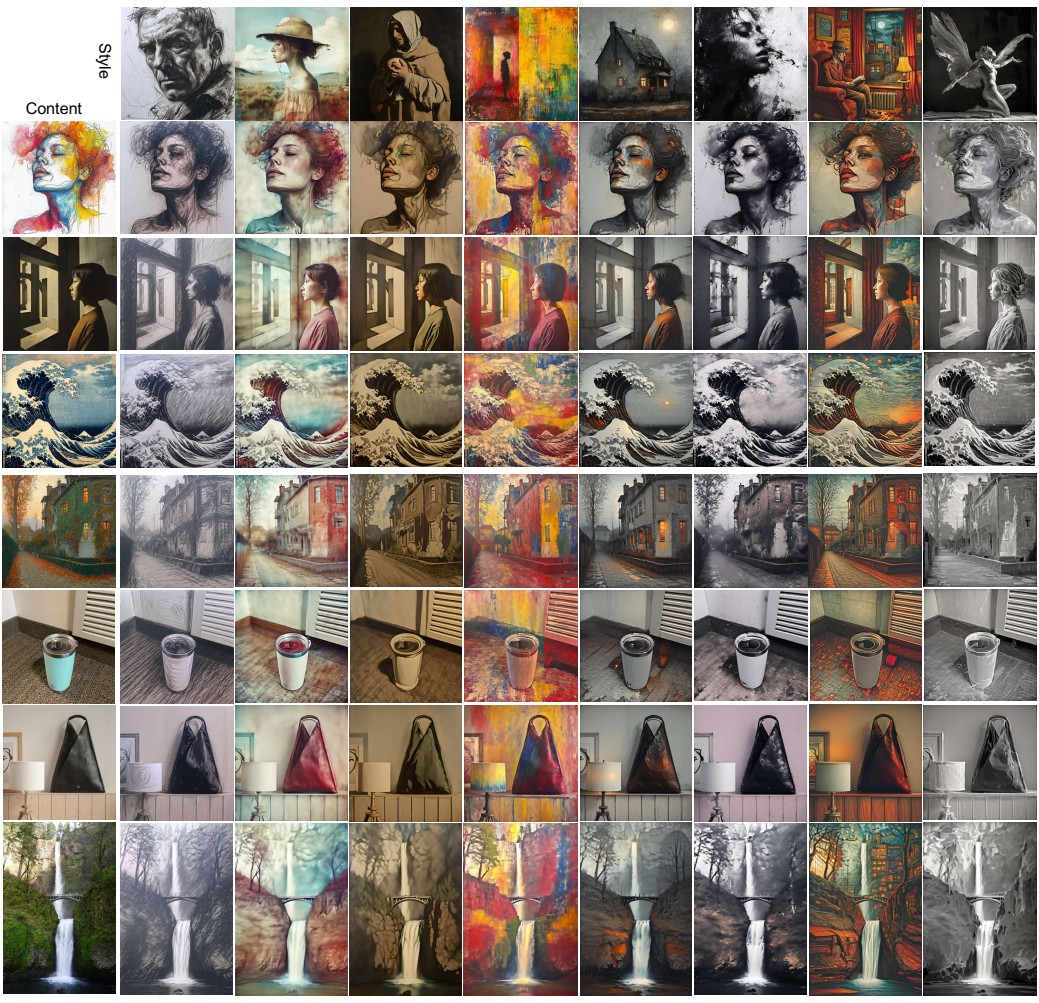

Figure 10: Image-driven style transfer results.

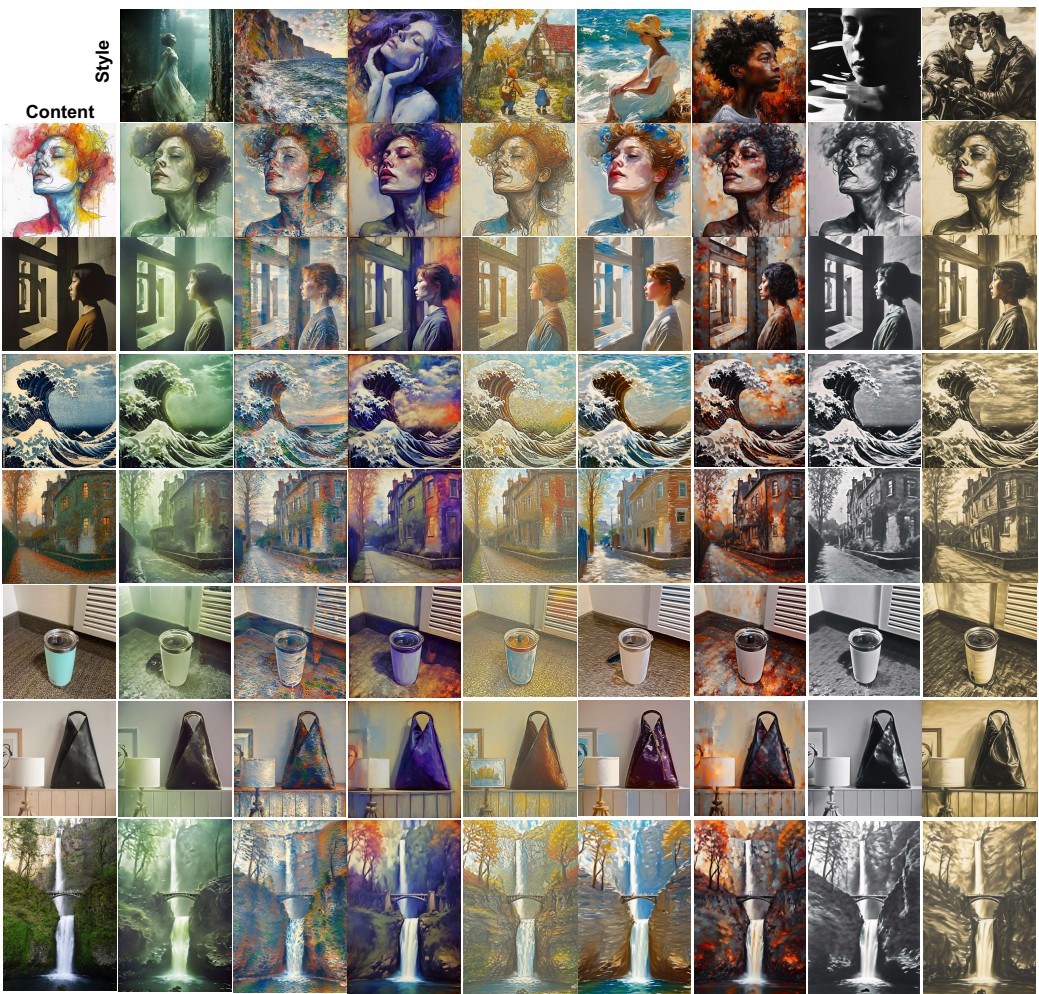

Figure 11: Image-driven style transfer results.

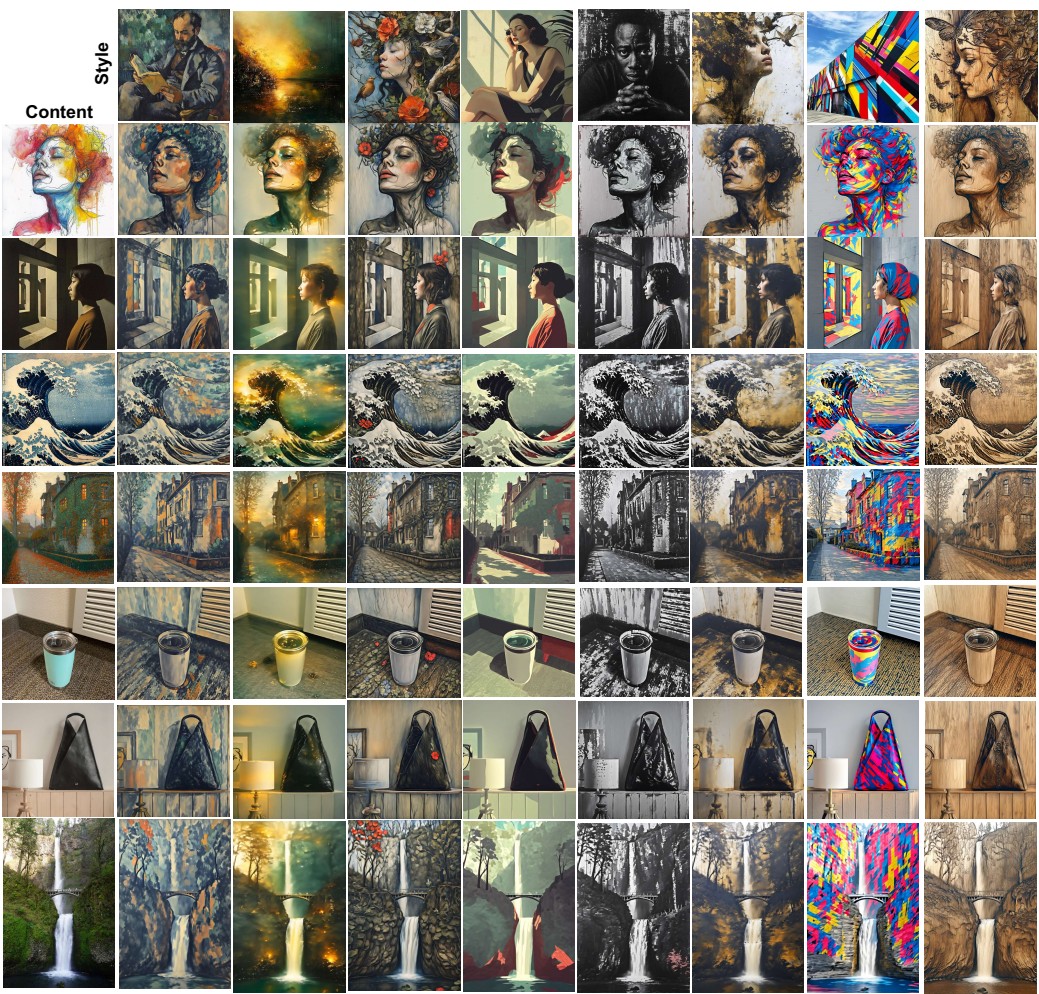

Figure 12: Image-driven style transfer results.

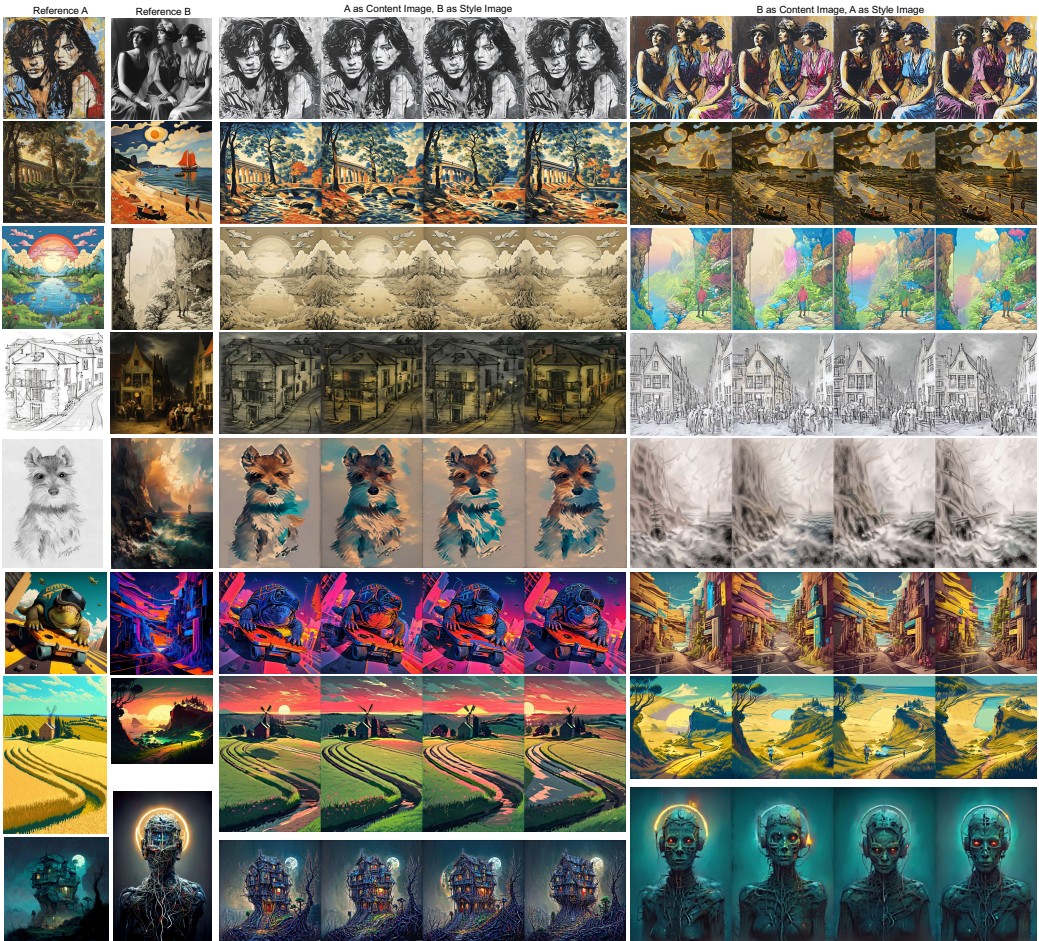

Figure 13: Content-style cycle transfer results.

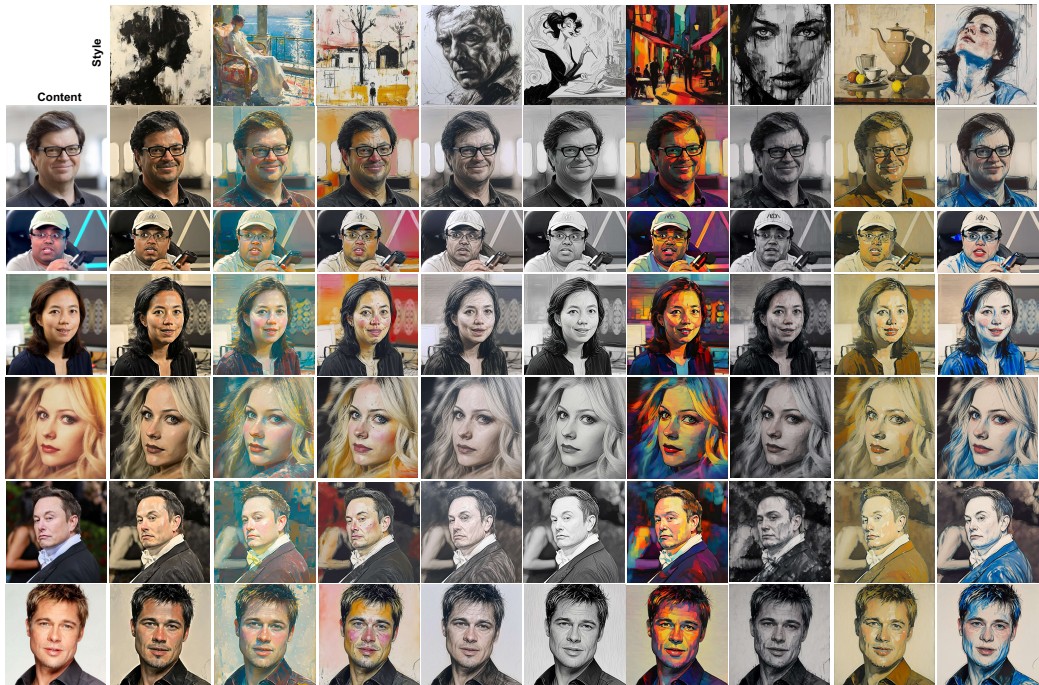

Figure 14: Face image style transfer results.

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

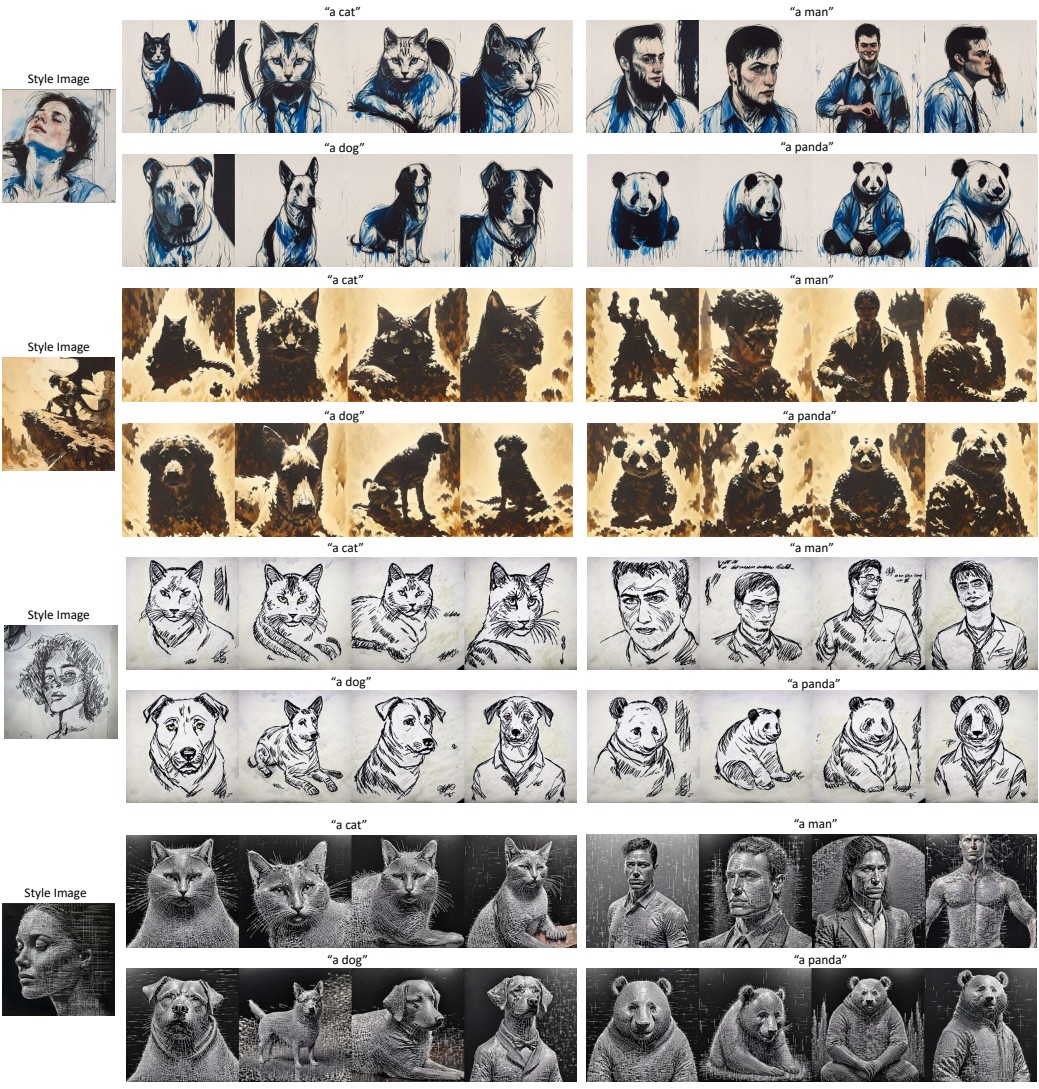

Figure 15: Text-driven stylized synthesis results.

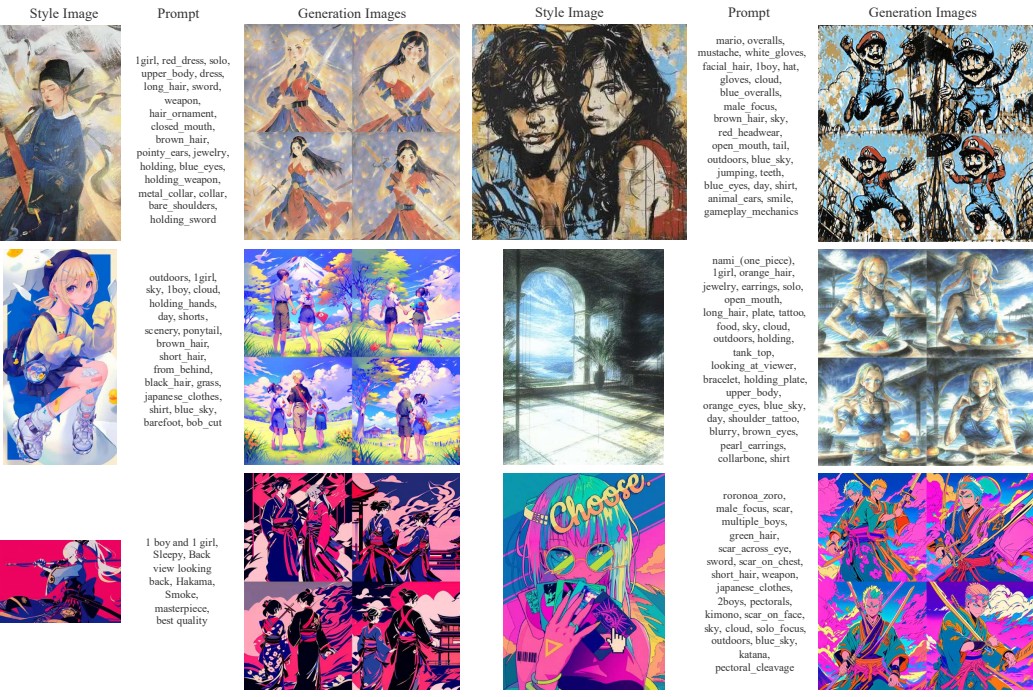

Figure 16: Text-driven stylized synthesis results.

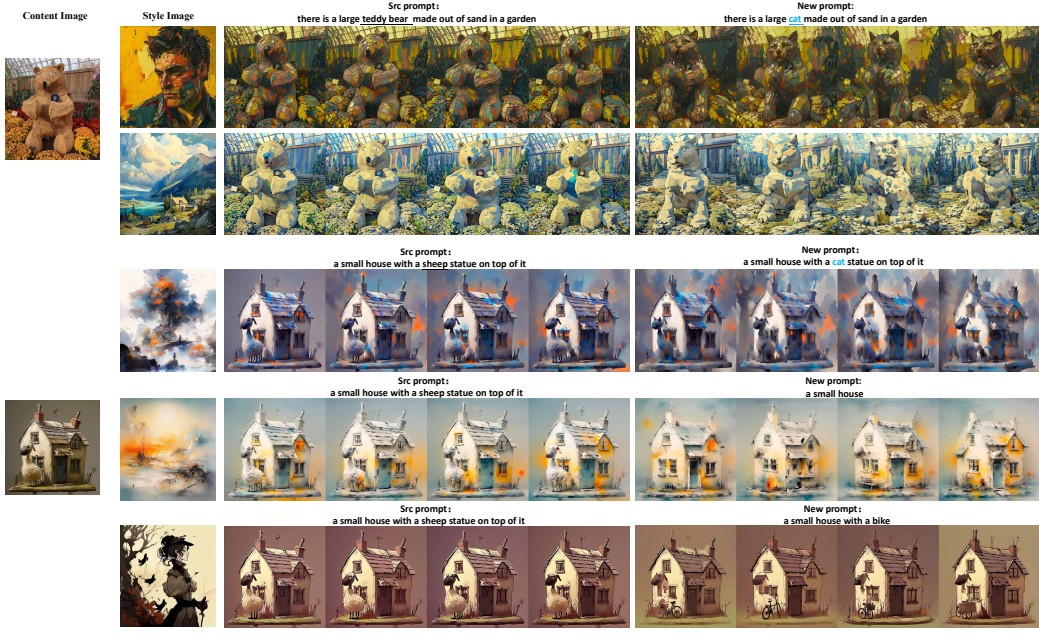

Figure 17: Text editing-driven stylized synthesis results.

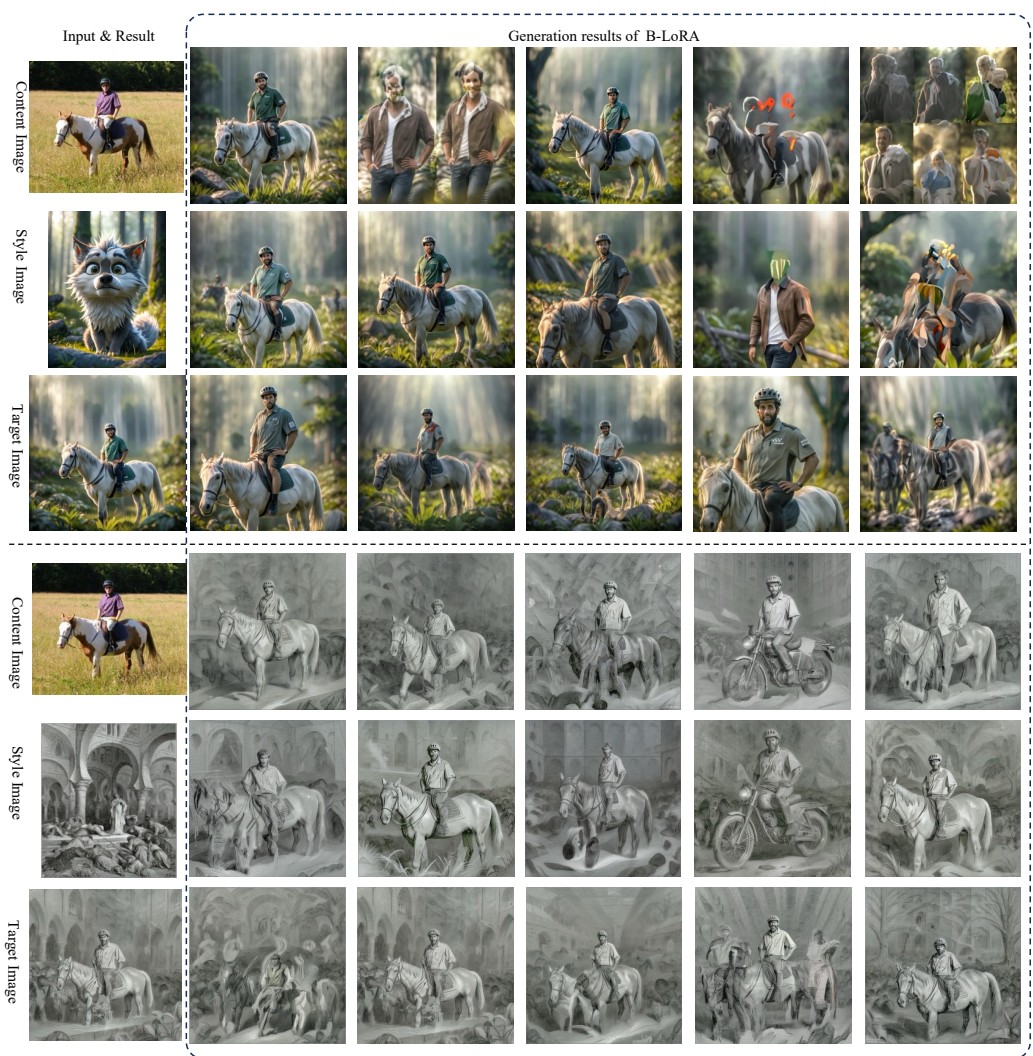

Figure 18: Example of data cleansing using CAS.

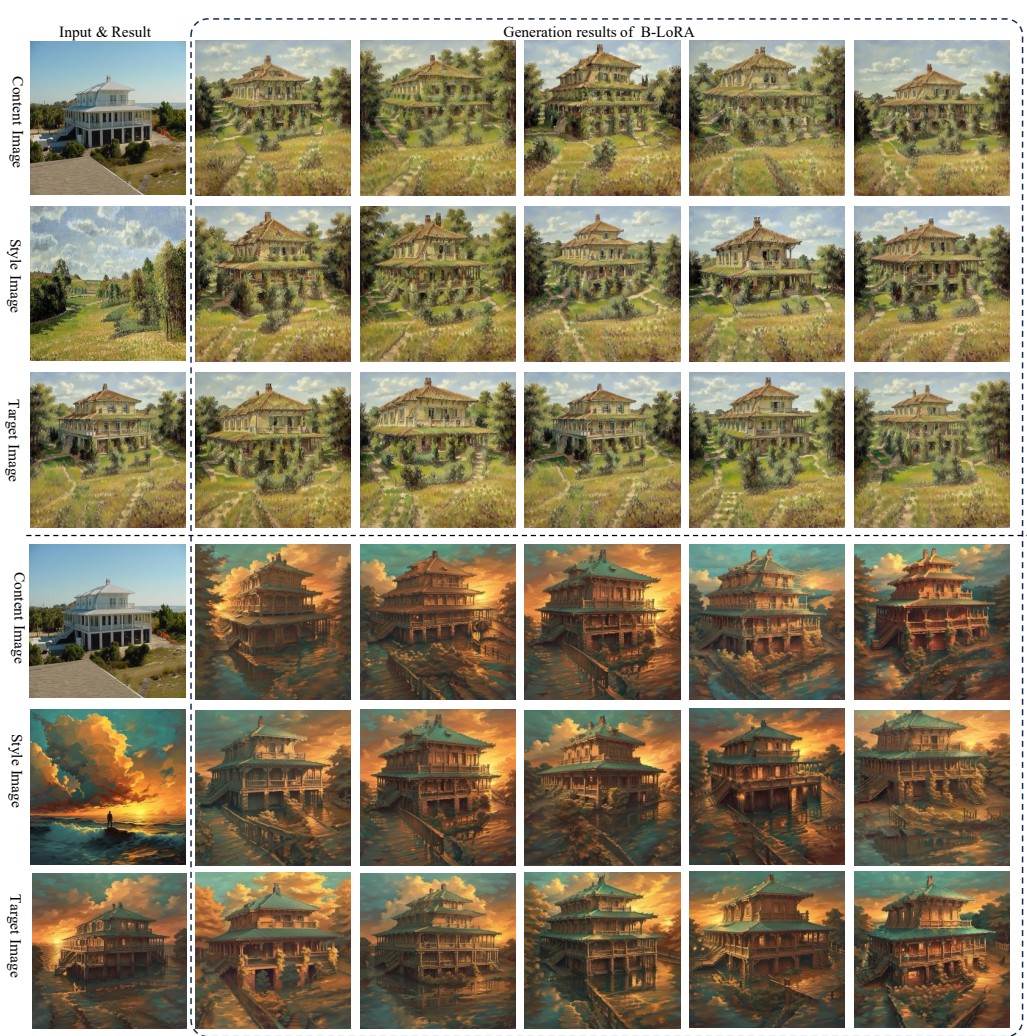

Figure 19: Example of data cleansing using CAS.

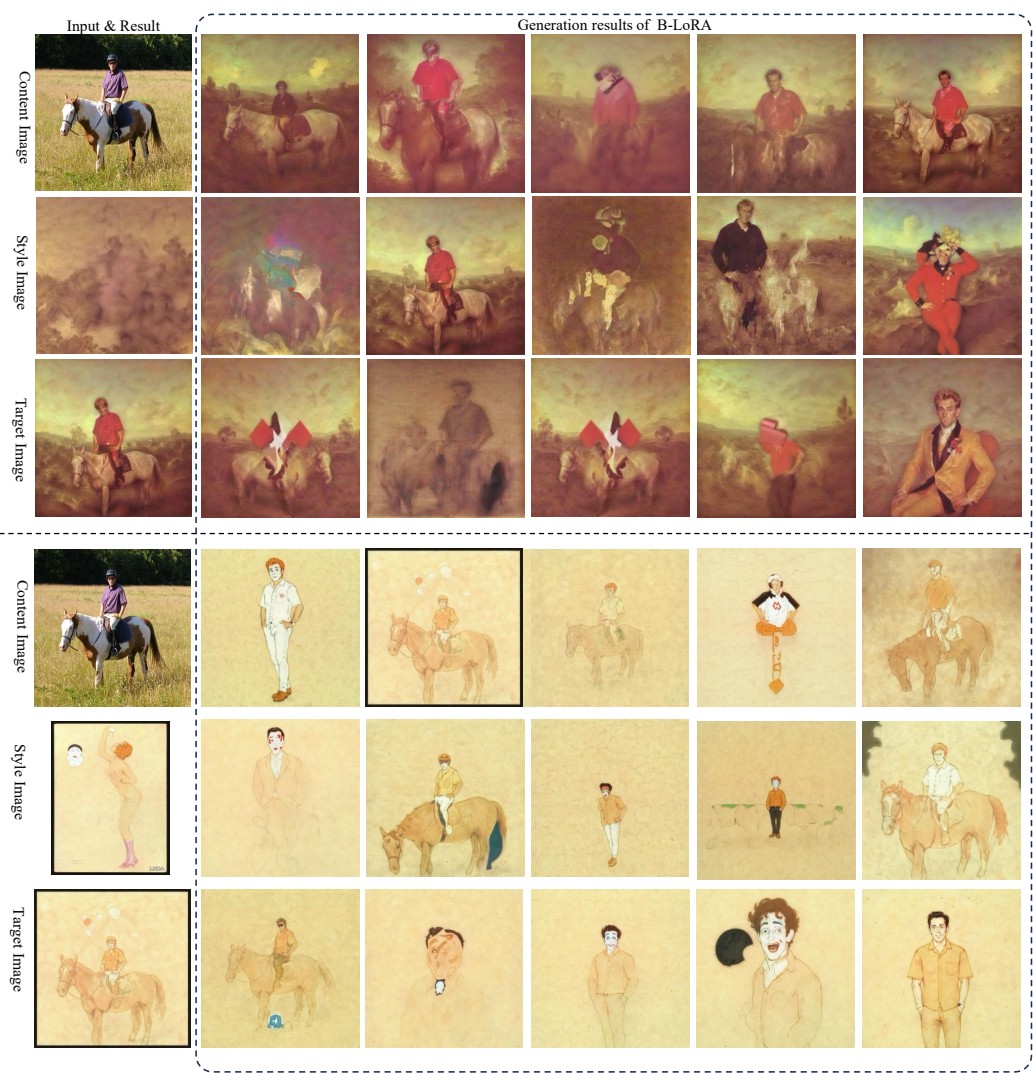

Figure 20: Example of data cleansing using CAS.

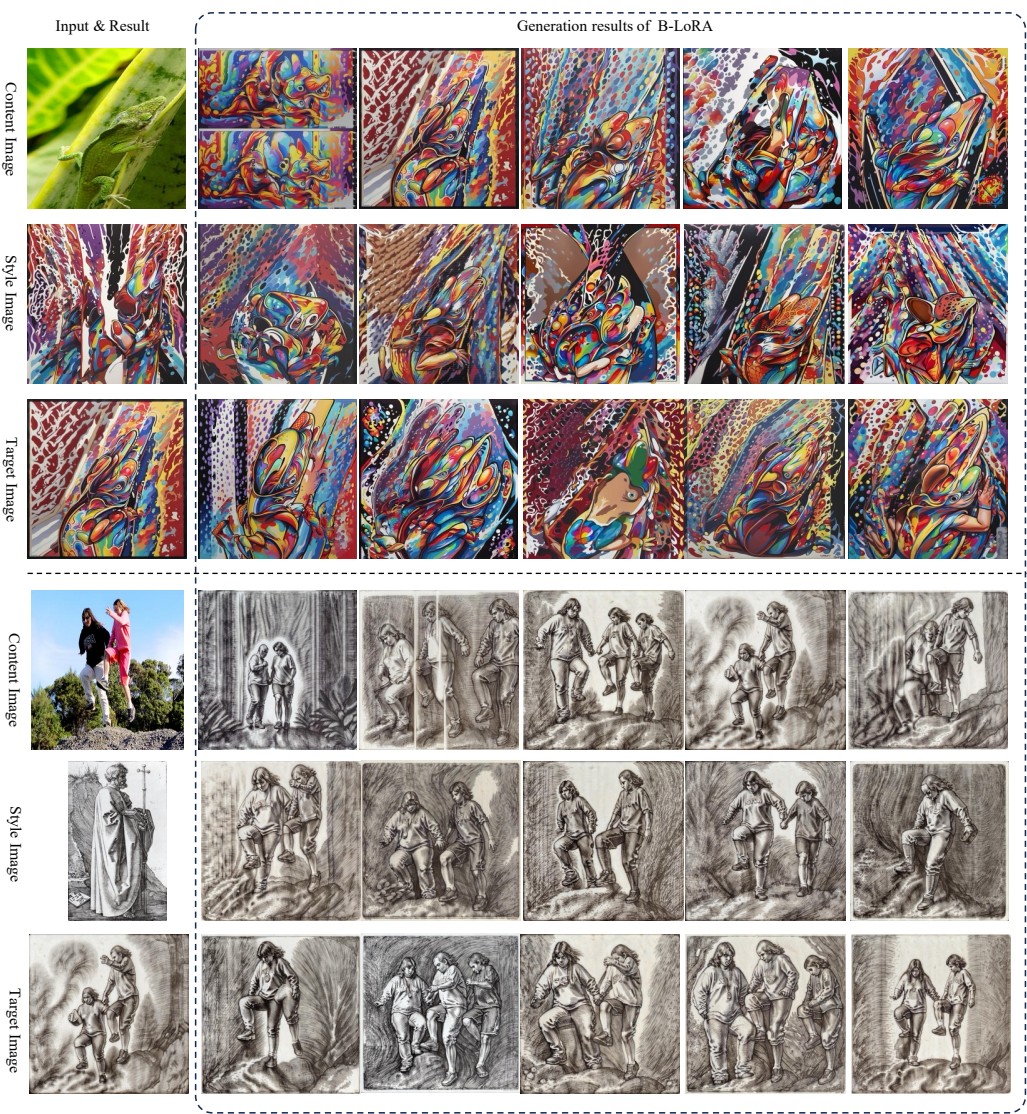

Figure 21: Example of data cleansing using CAS.

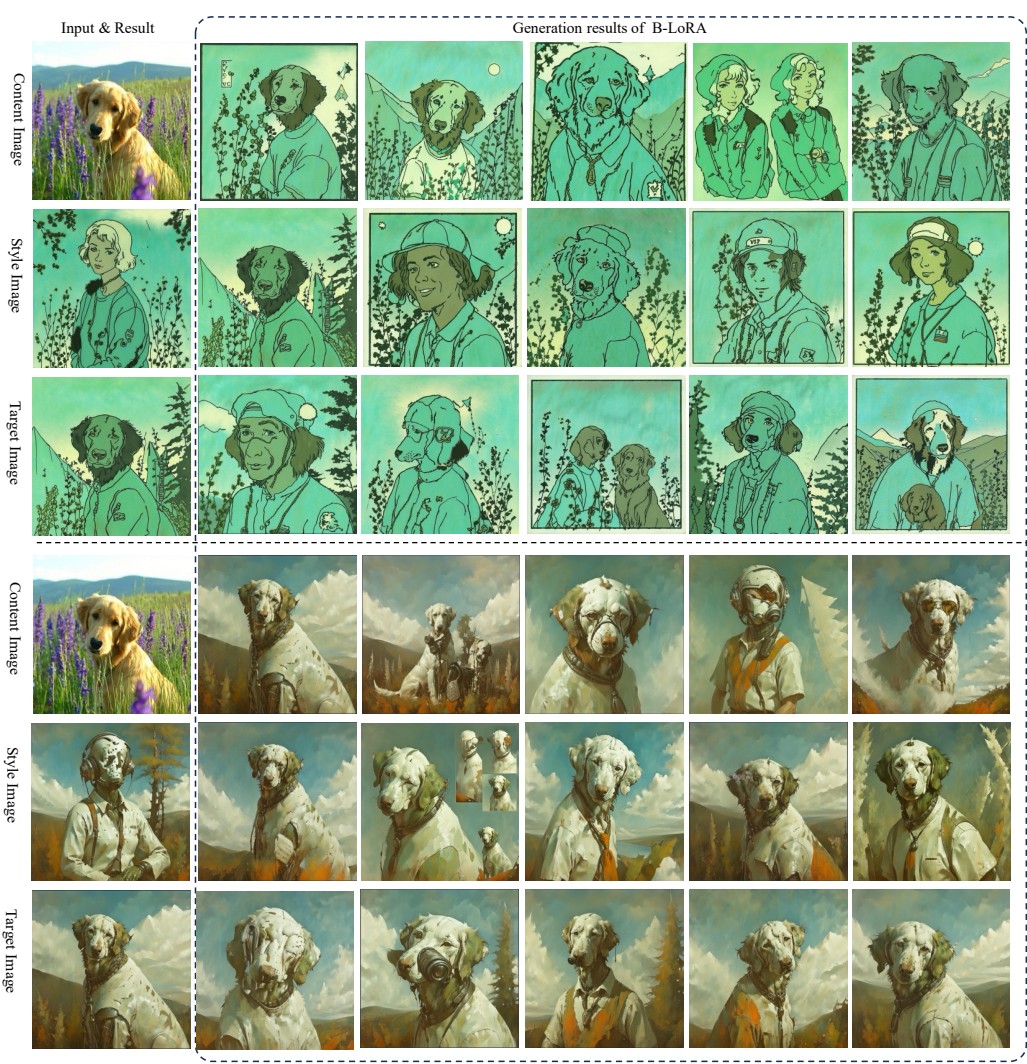

Figure 22: Example of data cleansing using CAS.