# OpenReview forum: "CSGO: Content-Style Composition in Text-to-Image Generation"
_ICLR.cc/2025/Conference — Submitted to ICLR 2025_

### Official Review · Reviewer_qQFs · 2024-10-29

**Soundness:** 2
**Presentation:** 2
**Contribution:** 2
**Rating:** 6
**Confidence:** 4

**Summary:**

The authors introduce a large dataset, IMAGStyle, consisting of 210k triplets, to train a style transfer model via a simple feature injection technique. To construct IMAGStyle, they collect arbitrary pairs of content and style images, (i) apply style transfer to the content images, and (ii) filter out stylized images that exhibit content leakage from the style images. They propose a straightforward adapter and controlnet based architecture with modification in cross attention and feature injection layers.

**Strengths:**

(i) The curated dataset, IMAGStyle, encompasses a broad range of content-style images, demonstrating extensive applicability.

(ii) The visual quality of the stylized image samples presented in the paper and appendix is highly satisfactory.

(iii) The concept is straightforward, and the methodology is direct.

**Weaknesses:**

(i) In the second step of data creation, AdaIN with DINO features is used to filter out images with style leakage. Does this process ensure the removal of style leakage related to factors such as content pose, size, and background? It is necessary to demonstrate that the filtering process addresses various types of style leakage, including texture, color, pose, size, and background.

(ii) The comparison with existing methods appears to omit several relevant works. There are more recent, well-performing approaches that utilize textual inversion variants, LoRA/DreamBooth variants, and training-free methods.

(iii) There are too many variants of hyper-parameters that affect the quality of image sample: content scale $\delta_c$, another content scale $\lambda_c$, style scale $\lambda_s$, and cfg scale. There needs to be specific hyper-parameter setting that generally leads to satisfactory results.

**Questions:**

(i) What are the primary differences between the proposed method and existing adapter variants, such as Ip-Adapter (Ye et al., 2023) and StyleAdapter (Wang et al., 2023), which leverage content-style pairs for adapter training? These models are noted for their limited generalizability to style images that were not included in the training data. Does CSGO exhibit the same limitation? Please provide examples using style images that include random cartoon characters not present in the WikiArt dataset.

(ii) In addition to qualitative samples and CSD similarity measurements, please include human evaluation results on randomly sampled stylized images.

---

> ### Author Response · Authors · 2024-11-21
> **Official Comment by Authors to Reviewer  qQFs(1/2)**
>
> Thank you for your recognition of our work and for your insightful comments.
>
> ---
>
> **Q1: In the second step of data creation, AdaIN with DINO features is used to filter out images with style leakage. Does this process ensure the removal of style leakage related to factors such as content pose, size, and background? It is necessary to demonstrate that the filtering process addresses various types of style leakage, including texture, color, pose, size, and background.**
>
> A: AdaIN features can be used to measure content information and style information. For the proposed dataset, we use **batch generation followed by cleaning** to ensure that the obtained triples are as accurate as possible. Specifically, for each content B-LoRA and style B-LoRA combination, we generate a total of **more than 50 images**. We then clean them using CAS metrics and use the image with the **smallest CAS score** as the target image. **We show examples in Figure 1 of the Supplementary Material showing that CAS can filter unreasonably generated images.** Meanwhile, Figure 2 of the original manuscript similarly shows examples. In Figure 3, we show the target image obtained by CAS metrics. In the experiment, the threshold is set to 1.6 and it can filter the failure cases of pose, size. In addition, for information such as color and texture, it can be filtered by calculating the style similarity through the CSD score [1]. On the other hand, the high quality style transfer results obtained from CSGO utilizing IMAGStyle will hopefully alleviate reviewers' concerns about dataset cleaning.
>
> [1]Gowthami Somepalli, Anubhav Gupta, Kamal Gupta, Shramay Palta, Micah Goldblum, Jonas Geiping, Abhinav Shrivastava, and Tom Goldstein. Measuring style similarity in diffusion models. In ECCV, 2024
>
> ---
>
> **Q2: The comparison with existing methods appears to omit several relevant works. There are more recent, well-performing approaches that utilize textual inversion variants, LoRA/DreamBooth variants, and training-free methods.**
>
> A: To the best of our knowledge, we have compared the current state-of-the-art style transfer methods StyleAlign[2], Styleshot[3], InstantStyle[4], StyleID[5].
> For text-driven stylized compositing tasks, in addition to the above methods we also compare with DEADiff[6].
> In addition, our approach is free of fine-tuning, presenting a significant advantage over LoRA approaches that continue fine-tuning for a set of style images.
> As suggested by the reviewers, we compare with the current SOTA style transfer LORA scheme B-LoRA. We trained 3 sets of B-LoRA and show the results in the Supplementary Material. It can be seen that CSGO, which does not require fine-tuning, outperforms the style transfer results of B-LoRA.
> It should be clarified that the main task of CSGO is style transfer, so the comparison method needs to achieve preservation of both content and style, whereas Dreambooth is only able to achieve preservation of one of them.
>
> [2]Amir Hertz, Andrey Voynov, Shlomi Fruchter, and Daniel Cohen-Or. Style aligned image generation via shared attention. In CVPR, 2024.
>
> [3]Gao Junyao, Liu Yanchen, Sun Yanan, Tang Yinhao, Zeng Yanhong, Chen Kai, and Zhao Cairong. Styleshot: A snapshot on any style. arXiv, 2024.
>
> [4]Haofan Wang, Qixun Wang, Xu Bai, Zekui Qin, and Anthony Chen. Instantstyle: Free lunch towards style-preserving in text-to-image generation. arXiv, 2024
>
> [5]Jiwoo Chung, Sangeek Hyun, and Jae-Pil Heo. Style injection in diffusion: A training-free approach for adapting large-scale diffusion models for style transfer. In CVPR, 2024.
>
> [6]Tianhao Qi, Shancheng Fang, Yanze Wu, Hongtao Xie, Jiawei Liu, Lang Chen, Qian He, and Yongdong Zhang. Deadiff: An efficient stylization diffusion model with disentangled representations. In CVPR, 2024.
>
> ---
>
> **Q3：hyper-parameters**
>
> A: As shown in Fig. 14 & Fig. 15 of the original paper, we show the results of the ablation experiments for the hyperparameters involved. In addition, we give the most suitable parameters for style transfer.

---

> > ### Comment · Reviewer_qQFs · 2024-11-22
> > **Clarification of the question**
> >
> > $\textbf{Q1: Dataset filtering process}$\\
> > I understand that the filtering step ensures that the best samples among 50 generations with CAS metric. I am curious if the CAS metric can truly filter out unreasonable generation that show diverse types of style leakage, \textit{including texture, color, pose, size, and background.} The response and figure 1 does not fully solve my concern if CAS is a robust metric to filter out the content information such as pose, size, and background coming from the style image. I suggest that the authors include extensive list of visual examples of filtering cases without cherry picking.

---

> > ### Author Response · Authors · 2024-11-28
> >
> > Dear Reviewer qQFs:
> >
> > As today is the last day to revise the manuscript, I wanted to kindly follow up regarding the concerns you raised earlier. We have already provided detailed responses to address your feedback, but we have not yet received any further comments or suggestions.
> >
> > If there are any remaining points or clarifications needed, please feel free to let us know. We greatly value your insights and are eager to ensure the final manuscript meets your expectations.
> >
> > Thank you for your time and thoughtful consideration.
> >
> > Best regards,
> >
> > CSGO Authors

---

> ### Author Response · Authors · 2024-11-21
> **Official Comment by Authors to Reviewer  qQFs(2/2)**
>
> **Q4：What are the primary differences between the proposed method and existing adapter variants, such as Ip-Adapter (Ye et al., 2023) and StyleAdapter (Wang et al., 2023), which leverage content-style pairs for adapter training?**
>
> A: None of them have usable pair data for training because of the difficulty of collecting images with the same set of styles in the previous work. Next, we will further describe the differences between CSGO and them.
> First of all, CSGO has a separate content control branch that can be utilized to more finely restrict the content of the original image. Therefore, CSGO can realize style transfer for image editing and style transfer for original image.
> However, IPAdapter and StyleAdapter only have single image control branch. For example, StyleAdapter can almost only control the content through text prompts, and lacks finer-grained control.
> Secondly, CSGO implements image-driven style transfer, text-driven stylized synthesis and text-editor-driven stylized synthesis at the same time. both IP-Adapter and StyleAdapter can only do text-driven stylized synthesis task.
> Finally, IPAdapter and StyleAdapter can only be trained by reconstruction way, i.e., the style image is the same as the target image, which leads to easy style leakage. The proposed CSGO utilizes the novel IMAGStyle dataset, where the control image and the target image only have the same style, which is more conducive to the learning of the style sampler. This further avoids leakage of style images.
>
> ---
>
> **Q5:These models are noted for their limited generalizability to style images that were not included in the training data. Does CSGO exhibit the same limitation? Please provide examples using style images that include random cartoon characters not present in the WikiArt dataset.**
>
> A:  When building the IMAGStyle dataset, we collected not only some of the images from the WikiArt dataset, but also style-generated images from Midjourney, which cover a wide range of styles. However, IPAdapter is not actually trained for style transfer . Therefore CSGO's style control is superior to IPAdapter and its variants.
>
> The style test sets presented in our manuscript and appendix are not all from WikiArt. e.g., the style image in Fig. 15 is from JourneyDB. the style images in Fig. 9, Fig. 10, Fig. 11, and Fig. 14 are randomly sampled from the web. Figures 4~18 in the Appendix are almost exclusively not from WikiArt; they are from Midjourney, JourneyDB, and online downloads.
>
> ---
>
> **Q6:human evaluation results**
> A: Thanks to the reviewer's suggestion, we added the human evaluation results.
> Setting: we randomly select 100 sets of results from the test set. Of these, 20 groups of portraits and 20 groups of sketches the rest were randomized. Subsequently, a user research experiment was conducted to compare CSGO with Styleshot-lineart, instantStyle, and Stylealigned respectively. Each group contains four generated results and the user selects the best result from the transfer quality.
> |VS|       CSGO win       | Tile | CSGO loss |
> |:----------------:|:----------------:|:-----------:|:-----------:|
> |StyleShot|     58.5%     |21.4%|20.1%|
> |Instanstyle|  64.2%   |20.6%|15.4%|
> |StyleAligned| 67.0% |12.3%|10.7%|
>
> ---
>
> **If our answers are more in line with your expectations, we kindly invite you to reconsider your initial rating.**

---

> ### Author Response · Authors · 2024-11-22
> **Response to Reviewer qQFs**
>
> A:We thank the reviewers for their feedback.
> First, the raw data was cleaned due to storage limitations. In response to the reviewers’ questions, we retrained multiple sets of B-LoRA. Subsequently, CAS cleaning was applied, **as shown in Figures 18,19, 20, 21, and 22 of the Supplementary Material**. In total, **10 sets** of generated images were filtered using CAS. These examples demonstrate that CAS effectively cleans illogical generated graphs in terms of pose, size, and other inconsistencies. However, we emphasize that since B-LoRA is inherently more stable for style generation, we rely on CSD for filtering these images. In our experiments, filtering with CAS alone is feasible without the need for CSD. For style images generated by arbitrary LoRA, we propose using CSA to compute style similarity and filter out images with inconsistent colors.
>
> We would like to show that CAS is very effective and intuitive for cleaning our style transfer data (comparing pixel-level differences after DINO feature removal style). Whether it is effective for the rest of the complex scenarios is something that needs to be further verified.

---

> ### Author Response · Authors · 2024-11-23
> **Follow-up**
>
> Dear Reviewer,
>
> We hope this message finds you well. We sincerely appreciate the time and effort you have dedicated to reviewing our submission. We have submitted our rebuttal and would like to follow up to inquire whether our responses have sufficiently addressed your concerns.
>
> Please let us know if you have any remaining questions or require additional clarification. We value your feedback and are eager to ensure our work meets the highest standards.
>
> Thank you again for your thoughtful insights and guidance.
>
> Best regards,
>
> CSGO Authors

---

> ### Comment · Reviewer_qQFs · 2024-11-27
>
> Thank you to the authors for addressing some of the issues.
>
> However, I still have reservations about the dataset filtering system using CAS.
>
> In the appendix, the authors appear to have omitted the CAS filtering results for some reason. Specifically, Figures ?? are not shown in the Appendix, as follows:
>
> > *we show 10 sets of CAS filtering examples in Figures ??. These cases show that CAS
> can clean illogical generated graphs for pose, size, and so on. However, we emphasize that since
> B-LoRA is actually more stable for the generation of styles, it is up to us to filter the images with
> CSD. In our experiments, it is possible to filter using only CAS without CSD.*
>
> I believe that the curation of a large-scale style-content dataset is a critical step in this area of research. Therefore, the dataset curation process should be presented in greater detail.
>
> As a result, the rating remains unchanged.

---

> > ### Author Response · Authors · 2024-11-27
> > **Response to Reviewer qQFs**
> >
> > Dear Reviewer qQFs
> >
> >
> > We have placed **Figures 18-Figure 22 at the end of the Appendix** showing the content image, the style image, and a large number of raw images generated by B-LoRA, respectively. In the bottom left corner of each figure we show the “target image”, which represents the best result obtained by CAS metrics. In addition, we have set a threshold filter, i.e., a high CAS score, which indicates a high level of content loss, will be filtered directly.
> >
> > I'm very sorry for the misunderstanding caused by the formatting error. We have re-corrected the citation format and re-uploaded the supplementary material. They are placed at the end of the supplementary material (Figure 18-figure  22).
> >
> > Best regards,
> >
> > CSGO Authors

---

> > > ### Comment · Reviewer_qQFs · 2024-11-27
> > >
> > > Thank you for the prompt response.
> > >
> > > It seems like the raw data of the curated style-content images is currently unavailable due to its removal from storage. I wonder if the dataset will be made available upon final submission. Is there an estimated timeline for re-generating the synthetic dataset?

---

> ### Author Response · Authors · 2024-11-27
> **Response to Reviewer qQFs**
>
> Dear Reviewer qQFs：
>
> We appreciate your thoughtful feedback. Here, we would like to provide clarifications to address some potential misunderstandings.
>
> Of course, the cleaned data, the IMAGStyle dataset, will certainly be published. What we are removing from storage is the raw data that has been generated directly by B-LoRA, and they are dirty data. It seems to us that these unprocessed data have no value. We will publish the IMAGStyle dataset obtained by CAS processing. Finally, if more style transfer data triples are needed, we can generate and then clean and filter them through the proposed CSGO, which is less costly and efficient compared to B-LoRA. We also hope that this framework and method can promote the development of style transfer.
>
> Best regards,
>
> CSGO Authors

---

### Official Review · Reviewer_y9qW · 2024-11-01

**Soundness:** 2
**Presentation:** 2
**Contribution:** 3
**Rating:** 5
**Confidence:** 5

**Summary:**

The paper establishes a data pipeline for constructing content-style-stylized image pairs and introduces the IMAGStyle dataset. Additionally, it utilizes this dataset to perform end-to-end training on the proposed CSGO framework, achieving style transfer generation under various input conditions.

**Strengths:**

- A high-quality dataset composed of content-style-stylized image pairs is proposed, which can be useful for research in style transfer.

- This paper proposes a method for decoupling content and style, injecting them separately into different locations within the U-Net. Additionally, it combines ControlNet for further integration of the injected features. In this approach, both U-Net and ControlNet have fixed parameters, creating an efficient training framework.

- The paper shows many analyses and visualization results of the proposed method, and easy to follow.

**Weaknesses:**

- If the dataset and its construction pipeline are one of the contributions of this paper, it is necessary to provide results using this dataset for training on other baseline methods and compare them with the proposed CSGO framework. This would demonstrate the effectiveness of the dataset and the robustness of the CSGO method.

- The proposed method borrows from IP-Adapter and ControlNet. And, the specific inputs for the three proposed Cross-Attention blocks and the method of feature injection have not been clearly explained.

- The quantitative evaluation is relatively limited, additional metrics could be included for assessment. On the other hand, regarding qualitative evaluation, the existing visual results do not intuitively reflect the advantages of this method. (Some additional evaluation metrics, such as FID, Aesthetic Scores, and user studies.)

- Figure 1 serves as the first illustration, and it should be clearly introduced, including the input image, output image, and comparison images, providing visual guidance to facilitate better understanding. This should also include the Figure 1(3) part.

- Suggestions for the format: citations need to be changed to conform more to the standard \citet or \citep; the font formatting and size of all similar-level figures in the paper are inconsistent, and the arrangement of images is rough and needs further improvement.

**Questions:**

- I am curious why the dataset is abbreviated as IMAGStyle and the method is abbreviated as CSGO.

- Can this method be applied to multiple content images or multiple style images as reference images to achieve better results?

- During the data cleaning phase, CAS is used to validate content consistency. How can we ensure that the style generated by LoRA for the images is correct?

- It appears to empirically inject Content into the down block and Style into the UpBlock; could you clarify the rationale behind this choice?

---

> ### Author Response · Authors · 2024-11-21
> **Official Comment by Authors to Reviewer y9qW(1/2)**
>
> Thanks to the reviewers for praising the high quality of the dataset and the methodology is easy to follow .
>
> ---
>
> **Q1：it is necessary to provide results using this dataset for training on other baseline methods and compare them with the proposed CSGO framework.**
>
> A: We thank the reviewers for their suggestions.
> We retrained the Stytr^2[1] model using IMAGStyle. In order to fully test the performance of IMAGStyle, we launched two experiments, the first is to retrain Stytr^2 using IMAGStyle only, and to  fine-tuning Stytr^2 using IMAGStyle (utilizing the released model weights of 160,000 steps) .
>
> Furthermore, since Stytr^2 employs a non-trivial training approach, instead it is implicitly constrained to generate results with content image close to the content image and style close to the style image. The significant advantage of IMAGStyle exists in the <content,style,target> triplet. Therefore, we add explicit pixel-level constraints to Stytr^2, which utilizes MSE loss to constrain the distance of the generated result from the target map in the triplet.
>
>
> |       Metric       |Stytr^2 | Stytr^2 on IMAGStyle|     Fine-tuning Stytr^2 on IMAGStyle       |CSGO|
> |:----------------:|:-----------:|:-----------:|:--------------------:|:--------------------:|
> |     CSD     |0.2695| 0.3430 |  0.3597   |0.5146|
> |  CAS   |0.9699|0.9332|   0.9280     |0.8386|
> | Aesthetics Score |4.0387|4.5146| 4.6975 | 5.5467|
>
>
> We retrained and fine-tuned the steps on the 8 A800-80G machines for 1W steps， batchsize is 24, and the results are shown in the table below. It can be clearly observed the effectiveness of IMAGStyle's triple data for the style transfer model.
>
>
> Stytr^2 utilizes the transformer framework for style transfer and has limited generative capabilities, unlike CSGO which utilizes a diffusion modeling scheme. In addition, CSGO is designed with separate content control and style control branches, which has advantages in controling content and style. More importantly, CSGO supports text-prompt-driven stylized synthesis and text-editor-driven stylized synthesis tasks, which Stytr^2 lacks.
>
>
> [1] Deng Y, Tang F, Dong W, et al. Stytr2: Image style transfer with transformers. Proceedings of the IEEE/CVF conference on computer vision and pattern recognition. 2022: 11326-11336.
>
> ---
>
> **Q2: Feature injection has not been clearly explained.**
>
> A: In the content control branch, we begin by extracting content features using the image encoder and passing them through the content mapping layer to obtain mapped content features. These features are represented as **key** and **value** matrices in the attention layers. The trans-attention layer features in the base UNet serve as the **query**, and the output features are fused with the original trans-attention layer input through pixel-level addition. A detailed explanation is provided in the Content Control section of the original manuscript.
>
> A similar approach is applied to the cross-attention layer in the style control branch. For ControlNet injection, the output features of each down_block and mid_block in controlnet are injected with the base model up_block and mid_block directly through element-level addition. We adopt the methodology outlined in [2], which is elaborated in the Content Control section of the manuscript and further detailed in the appendix.
>
> [2] Zhang, Lvmin, Anyi Rao, and Maneesh Agrawala. "Adding conditional control to text-to-image diffusion models." Proceedings of the IEEE/CVF International Conference on Computer Vision. 2023.
>
> ---
>
> **Q3: FID, Aesthetic Scores**
>
> A:
>
> |Metric |Stytr^2|Style-Aligned | StyleID|     InstantStyle       |StyleShot|StyleShot-lineart|CSGO|
> |:----------------:|:-----------:|:-----------:|:--------------------:|:-----------:|:-----------:|:--------------------:|--------------------:|
> FID|3.2729|2.5732|5.1680|2.6308|2.2395|2.1694|2.0391|
> Aesthetics Score|4.0387|3.7463|4.7643|5.4824|5.6728|5.2542|5.5467|
>
>
> We thank the reviewers for their valuable suggestions and agree that the inclusion of additional metrics is warranted. As a result, we have provided further quantitative metrics, which are presented in the table below. Intuitively, when the content and style do not blend well, the aesthetic score tends to be lower, and the FID (Fréchet Inception Distance) will also decrease. Based on these observations, we believe that CSGO remains a highly competitive model for style transfer.
>
> ---

---

> > ### Author Response · Authors · 2024-11-21
> > **Official Comment by Authors to Reviewer y9qW(2/2)**
> >
> > **Q4:user study**
> >
> > A: Thanks to the reviewer's suggestion, we added the human evaluation results.
> > Setting: we randomly select 100 sets of results from the test set. Of these, 20 groups of portraits and 20 groups of sketches the rest were randomized. Subsequently, a user research experiment was conducted to compare CSGO with Styleshot-lineart, instantStyle, and Stylealigned respectively. Each group contains four generated results and the user selects the best result from the transfer quality.
> >
> >
> > |VS|       CSGO win       | Tile | CSGO loss |
> > |:----------------:|:----------------:|:-----------:|:-----------:|
> > |StyleShot|     58.5%     |21.4%|20.1%|
> > |Instanstyle|  64.2%   |20.6%|15.4%|
> > |StyleAligned| 67.0% |12.3%|10.7%|
> >
> >
> > ---
> >
> > **Q5: Figure 1 serves as the first illustration, and it should be clearly introduced, including the input image, output image, and comparison images, providing visual guidance to facilitate better understanding. This should also include the Figure 1(3) part.**
> >
> > A:  Thanks to the reviewers' suggestion, we labeled inputs and outputs in the original manuscript.  For the image-driven stylistic transfer task, the content image and the style image serve as the two main inputs, and the diffusion model outputs the target image (shown in Fig. 1(1)). For the text-driven stylized synthesis task, the style image and the driven text prompts are the main inputs, and the content image can be set to null value, which can get the results in Figure 1(2). For the text editting-driven stylized synthesis task, given the edited text, the content image and the style image as inputs, the results in Figure 1(3) can be obtained.
> >
> > ---
> >
> > **Q6:format**
> > A: Thanks to the reviewers' suggestions, we are already actively revising the formatting of the original manuscript.
> >
> > ---
> >
> > **Q7: datase name and method name**
> > A: The dataset names and method names draw on style, content and style keywords, and also just pronouns.
> >
> > **Q8: Can this method be applied to multiple content images or multiple style images as reference images to achieve better results?**
> >
> > A: With the current version, CSGO cannot support multiple content images and multiple style images. Also multiple content images as input may be ambiguous for the style transfer task. In the future, we will continue to develop work that maintains consistent content image subject relationships and styles. We believe multiple content and multiple style images will be helpful in this task.
> >
> >
> > **Q9 :During the data cleaning phase, CAS is used to validate content consistency. How can we ensure that the style generated by LoRA for the images is correct?**
> >
> > A: First, we ensure style consistency based on B-LoRA[1]. In addition, for each style, we batch generated at least 50 images using the B-LoRA obtained from training, and then filtered them by CSD score (CSD>0.7). Finally, the stylized result with the highest score was used as the target image.
> > [1]Yarden Frenkel, Yael Vinker, Ariel Shamir, and Daniel Cohen-Or. Implicit style-content separation using b-lora. ECCV, 2024.
> >
> > **Q10：It appears to empirically inject Content into the down block and Style into the UpBlock; could you clarify the rationale behind this choice?**
> >
> > A: The design of CSGO's Content Block and Style Block is informed solely by experimental results and the conclusions drawn from InstantStyle. According to InstantStyle, the Style Block is located in the up_blocks.0.attentions.1 layer, although it relies on the original weights of IPAdapter. Consequently, we first validated the effectiveness of up_blocks.0.attentions.1 for style control. Subsequently, we incrementally added additional layers to evaluate the extent of the style transfer capability.
> >
> > The experimental results are presented in the table below. It is important to note that these results represent early experimental validation and do not involve the use of ControlNet to regulate content. Instead, only Content Blocks and cross-attention layers were utilized to manage content. The Content Block is applied to all blocks except those designated as Style Blocks.
> >
> >
> > |style block|CSD|
> > |:-----------:|:-----------:|
> > |up_blocks.0.attentions.1|0.5239|
> > |up_blocks.0.attentions.1 & up_blocks.0.attentions.2|0.5527|
> > |up_blocks.0.attentions.1 & up_blocks.0.attentions.2 & up_blocks.1|0.5743|
> > |up_blocks|0.5864|
> > |up_blocks & mid_blocks|0.5702|
> >
> >
> > The experimental results show that higher CSD scores can be obtained when setting up_blocks to style blocks. In addition, we also investigated the setting of overlapping content block and style block. However, we found that this may cause severe conflicts and significantly reduce the style transfer capability. Therefore, we employ decoupled controls.
> >
> > ---
> >
> > **If our answers are more in line with your expectations, we kindly invite you to reconsider your initial rating.**

---

> ### Author Response · Authors · 2024-11-23
> **Follow-up**
>
> Dear Reviewer,
>
> We hope this message finds you well. We sincerely appreciate the time and effort you have dedicated to reviewing our submission. We have submitted our rebuttal and would like to follow up to inquire whether our responses have sufficiently addressed your concerns.
>
> Please let us know if you have any remaining questions or require additional clarification. We value your feedback and are eager to ensure our work meets the highest standards.
>
> Thank you again for your thoughtful insights and guidance.
>
> Best regards,
> CSGO Authors

---

> > ### Comment · Reviewer_y9qW · 2024-11-26
> >
> > Thank you to the author for the reply. The author's response has addressed most of my concerns regarding the paper and has helped enhance its overall completeness. I will adjust my rating accordingly.
> > However, I still have some reservations about the contribution of the method and the formatting issues in the revised version.

---

> > > ### Author Response · Authors · 2024-11-26
> > > **Response to Reviewer y9qW**
> > >
> > > Dear Reviewer y9qW，
> > >
> > > Thank you for raising the score! We sincerely appreciate your recognition of our work and your valuable feedback.
> > >
> > > In particular, thanks to the reviewer's reminder, we carefully checked and revised the details of the images.
> > > Including but not limited to the following parts:
> > >
> > > 1)  adjusted the arrangement of subfigure 2 of Fig. 1 to make it more reasonable.
> > >
> > > 2) Aligned the image alignment of Fig. 2, Fig. 5, Fig. 6, Fig. 7, Fig. 9, Fig. 10, and Fig. 11.
> > >
> > > 3) Aligned the text size in Figures 6, 7 and 8.
> > >
> > > 4) Fixed formatting issues with citep and citet.
> > >
> > > Please let us know if further clarifications are needed or if there are any remaining points you would like us to address. We greatly value your feedback and are eager to work towards resolving all concerns. We will continue refining this method and hope to contribute even more impactful work in the future.
> > >
> > > Thank you again for your time and thoughtful input.
> > >
> > > Best regards,
> > >
> > > CSGO Authors

---

### Official Review · Reviewer_zBGr · 2024-11-03

**Soundness:** 3
**Presentation:** 3
**Contribution:** 2
**Rating:** 5
**Confidence:** 5

**Summary:**

This study proposes a diffusion-based stylized image generation method. The authors claim that the lack of paired data for training models limits the performance of popular stylization methods. To this end, the authors propose an augmented dataset by training various LoRAs for different contents and styles. Then, an IP-adaptor style framework is trained on the collected dataset.

**Strengths:**

Overall, the reviewer feels the motivation is valid. Given that many image-generation tasks are ill-posed and lacking ground truth, trying to find the paired data for supervised learning is valid. Also, the reviewer would like to express appreciation for the efforts in collecting the dataset, which may be quite time-consuming. The proposed CSGO is reasonable and easy to follow.

**Weaknesses:**

- The authors claim that the performance of image style transfer is limited because of the lack of a large-scale stylized dataset, which makes it impossible to train modes end-to-end. However, the proposed dataset is learned by training and combining different LoRAs, which means the generated stylized data is not the real ground truth for end-to-end training. In fact, the whole framework seems to try to distill the generated dataset in one adaptor.
- Most image generation tasks are ill-posed and lack ground truth. A similar idea goes to [r1] ''Identity-Preserving Face Swapping via Dual Surrogate Generative Models.''  Face-swapping methods try to fuse one source image with one target image. Similar to the setting of image style transfer, no ground truth information could be collected as image style transfer tasks for face-swapping tasks.  Thus, the authors of [r1] tried to generate the <source, target, results> triplets. A more careful analysis of the pros and cons of using such generated data is given in [r1]. However, in this study, the author claims that style transfer lacks a larger-scale stylized dataset without careful analysis or support.
The proposed method, CSGO, is just a combination of many existing techniques. For content control, the two strategies are the combination of ControlNet and IP-Adaptor. For style control, they employ Perceiver Resampler structure as Alayrac et al. to project the style features and then do some trivial modifications to controller or ip-adaptor. The reviewer understands that the authors need to verify the usefulness of the proposed dataset. However, such a method could not make a good contribution to ICLR.
- The proposed evaluation index has the same problem. Using AdaIN for content and style evaluation is common sense in a related field, but it could not be counted as a contribution.
- No user study is conducted. Technical writing should be paid more attention. For example, most of the cross-reference format is wrong (Maybe there is a mistake in using cite citet and citep)

**Questions:**

Could the collected dataset supplement the training of traditional style transfer methods? For example, the collected pair can be used to tune Stytr or CAST. Using the collected dataset and the training pipeline of traditional style transfer tasks, could the method perform much better than before?

---

> ### Author Response · Authors · 2024-11-21
> **Official Comment by Authors to Reviewer zBGr（1/2）**
>
> We thank the reviewers for recognizing the **motivation is valid** of our work.
>
> ---
>
> **Q1: The authors claim that the performance of image style transfer is limited because of the lack of a large-scale stylized dataset, which makes it impossible to train modes end-to-end. However, the proposed dataset is learned by training and combining different LoRAs, which means the generated stylized data is not the real ground truth for end-to-end training. In fact, the whole framework seems to try to distill the generated dataset in one adaptor.**
>
> A: We would like to clarify that a triad training approach, which uses generated target images alongside real content and style images, is closer to end-to-end training compared to a two-stage approach. For the style transfer task, it is almost impossible to collect sufficient <content, style, stylization result> data of the real scenario.
> Second, the proposed IMAGStyle is obtained through batch generation and high quality cleaning. Even though there are some limitations in the images generated by lora, the triples we get after filtering and manually checking by CSD and CAS metrics are in line with human intuition. This approach is also cost-effective, with a single B-LoRA achieving optimal performance within approximately 10 minutes on H800 or 15minutes on V100. Using the proposed CSGO framework, we have verified that the generated triples support high-quality style transfer.
> Finally, we would like to highlight that the CSGO framework effectively decouples the content and style control branches. Specifically, we use adapters to inject style features into ControlNet and the base model, respectively. Once trained, the CSGO framework unifies three types of stylistic control tasks: content-image-driven style transfer, text-driven stylized synthesis, and text-editor-driven stylized synthesis, as illustrated in Fig. 1.
>
> ---
>
> **Q2: Discusses relationship to face-swapping work [r1] and clarifies CSGO contributions**
>
> A: We thank the reviewer for the introduction and appreciate their observations. We agree that our approach does not conflict with the face-swapping scheme using <source, target, results>. Work [r1] utilizes real images to construct the source and target images in <Source, Target, Result>, which ensures that the target learned by the model comes from the real scene. However, the difference between the face-swapping task and the style transfer task is that it is difficult to construct content images and style images from real images. Style transfer involves both high- and low-dimensional features, such as color, texture, hue, and strokes. Although we can apply some image data-augmentation schemes to fade to other images, it limits the diversity of style transfer. For instance, it was challenging to degrade a furry doll as a target image into other styles as fake content images. Interestingly, our early approach was to generate fake content and style images by fading the style images. However, the results are significantly less effective than using the proposed IMAGStyle dataset. Therefore, it is a feasible way to construct the target image by real content image and real style image.
>
> We believe that ControlNet and IPAdapter have become the most effective and widely accepted methods for feature injection in the era of diffusion modeling. They offer simplicity and reliability, making them ideal for scenarios requiring feature injection. Furthermore, we emphasize that the main contribution of this paper is to provide a set of style transfer dataset construction and cleaning methods while annotating a high-quality segmentation migration dataset. With the support of this dataset, we utilize the mainstream framework to build a simple but effective style transfer framework, CSGO, which enables CSGO to unify three key style control tasks: image-driven style transfer, text-driven stylized synthesis, and text editing-driven stylized synthesis tasks through independent content control and style control.
>
> [r1] Huang, Ziyao, et al. "Identity-Preserving Face Swapping via Dual Surrogate Generative Models." ACM Transactions on Graphics 43.5 (2024): 1-19.
>
> ---

---

> > ### Author Response · Authors · 2024-11-21
> > **Official Comment by Authors to Reviewer zBGr（2/2）**
> >
> > **Q3: The proposed evaluation index has the same problem. Using AdaIN for content and style evaluation is common sense in a related field, but it could not be counted as a contribution.**
> >
> > A: AdaIN features of the DINO model is an important metric when cleaning our data, it is not our main contribution. Our primary contribution is the development of the IMAGStyle ternary dataset construction and cleaning method. Additionally, we propose a simple yet effective CSGO framework that decouples content and style features while uniformly implementing three types of style control tasks: image-driven style transfer, text-driven stylized synthesis, and text editing-driven stylized synthesis tasks.
> >
> > ---
> >
> > **Q4: User study**
> > A: Setting: we randomly select 100 sets of results from the test set. Of these, 20 groups of portraits and 20 groups of sketches the rest were randomized. Subsequently, a user research experiment was conducted to compare CSGO with Styleshot-lineart, instantStyle, and Stylealigned respectively. Each group contains four generated results and the user selects the best result from the transfer quality.
> > |VS|       CSGO win       | Tile | CSGO loss |
> > |:----------------:|:----------------:|:-----------:|:-----------:|
> > |StyleShot|     58.5%     |21.4%|20.1%|
> > |Instanstyle|  64.2%   |20.6%|15.4%|
> > |StyleAligned| 67.0% |12.3%|10.7%|
> >
> > ---
> >
> > **Q5：Format**
> >
> > A: We thank the reviewers for their careful review of the original manuscript, which we have carefully revised.
> >
> >
> > ---
> >
> > **Q6: Could the collected dataset supplement the training of traditional style transfer methods?**
> >
> > A: We thank the reviewers for their valuable suggestions. In response, we retrained the StyTr^2 [1] model using the IMAGStyle dataset. To comprehensively evaluate the performance of IMAGStyle, we conducted two experiments. First, we retrained StyTr^2 using only IMAGStyle. Second, we fine-tuned StyTr^2 using IMAGStyle, leveraging the released model weights pre-trained for 160,000 steps.
> > StyTr^2 employs a non-trivial training approach, wherein the model is implicitly constrained to produce results with content closely aligned to the content image and style closely aligned to the style image. The primary advantage of IMAGStyle lies in its <content, style, target> triplet structure. To further enhance the performance of StyTr^2, we introduced explicit pixel-level constraints by incorporating MSE loss. This addition enforces the generated results to be closer to the target map within the triplet, thereby improving style transfer fidelity.
> >
> >
> > |       Metric       |Stytr^2 | Stytr^2 on IMAGStyle|     Fine-tuning Stytr^2 on IMAGStyle       |
> > |:----------------:|:-----------:|:-----------:|:--------------------:|
> > |     CSD     |0.2695| 0.3430 |  0.3597   |
> > |  CAS   |0.9699|0.9332|   0.9280     |
> > | Aesthetics Score |4.1387|4.5146| 4.6975 |
> >
> >
> > We retrained and fine-tuned the steps on the 8 A800-80G machines for 1W steps， batchsize is 24, and the results are shown in the table below. It can be clearly observed the effectiveness of IMAGStyle's triple data for the style transfer model.
> >
> > [1] Deng Y, Tang F, Dong W, et al. Stytr2: Image style transfer with transformers. Proceedings of the IEEE/CVF conference on computer vision and pattern recognition. 2022: 11326-11336.
> >
> > ---
> >
> > **If our answers are more in line with your expectations, we kindly invite you to reconsider your initial rating.**

---

> ### Author Response · Authors · 2024-11-23
> **Follow-up**
>
> Dear Reviewer,
>
> We hope this message finds you well. We sincerely appreciate the time and effort you have dedicated to reviewing our submission. We have submitted our rebuttal and would like to follow up to inquire whether our responses have sufficiently addressed your concerns.
>
> Please let us know if you have any remaining questions or require additional clarification. We value your feedback and are eager to ensure our work meets the highest standards.
>
> Thank you again for your thoughtful insights and guidance.
>
> Best regards,
> CSGO Authors

---

> ### Author Response · Authors · 2024-11-28
>
> Dear Reviewer zBGr:
>
> As today is the last day to revise the manuscript, I wanted to kindly follow up regarding the concerns you raised earlier. We have already provided detailed responses to address your feedback, but we have not yet received any further comments or suggestions.
>
> If there are any remaining points or clarifications needed, please feel free to let us know. We greatly value your insights and are eager to ensure the final manuscript meets your expectations.
>
> Thank you for your time and thoughtful consideration.
>
> Best regards,
>
> CSGO Authors

---

### Official Review · Reviewer_Vjbz · 2024-11-04

**Soundness:** 3
**Presentation:** 3
**Contribution:** 3
**Rating:** 6
**Confidence:** 4

**Summary:**

This paper presents a method for reference-based image stylization method. To achieve this goal, a style encoder and an image encoder are presented. Features are injected into the diffusion backbone through selected layers. Meanwhile, a 210K (image, style, stylized image) triplet dataset was built and used for training the proposed model. In the experimental results, content similarity and style similarity were evaluated. Comparisons were made between the proposed method and other SOTA methods. Ablation studies are informative and comprehensive.

**Strengths:**

1. The paper is pretty complete, including a solid dataset, training details, evaluation comparisons and ablation studies.
2. The proposed dataset is useful for future related work

**Weaknesses:**

1. The overall training method is not new, which follows the lines of research like IP-adapter, InstanceID, InstanceStyle, and the usage of Ada-In is common too in stylization research.
2. In the dataset, the stylized image is fixed while given a content image and style image, a stylized image can be various based on preference. Since the extent to which the content image is stylized is fixed, the proposed network fits to that, which reduced the potential capacity to adapt. I understand there's control factor to adjust to be more stylized or less, but the upper bound is the dataset.

**Questions:**

Can you show some failure cases?

---

> ### Author Response · Authors · 2024-11-21
> **Official Comment by Authors to Reviewer Vjbz**
>
> We thank the reviewers for recognizing the **pretty complete**, **solid dataset**, and **useful** of our article.
>
> ---
>
> **Q1: Difference of CSGO with recent research like IP-adapter, InstanceID, InstanceStyle, and the usage of Ada-In is common too in stylization research.**
>
> A: Our main contributions include the introduction of a construction and cleaning method for constructing style migration datasets, and the labeling of the first high-quality style transfer dataset, IMAGStyle, which opens up the possibility of end-to-end style transfer. Additionally, we propose a simple and efficient CSGO unified framework following the mainstream frameworks to verify that the constructed and cleaned datasets are enhanced for style transfer. Essentially, the main role of the CSGO framework is to validate the dataset as well as unify multiple style transfer tasks. With the help of IMAGStyle, the simple CSGO framework can unify image-driven style transfer, text-driven stylized synthesis, and text-editor-driven stylized synthesis tasks.
> AdaIN is the metric we use to determine content similarity in data cleaning tasks, which is very effective.
>
> Regarding the reviewer’s comment, our current approach involves injecting style features using IPAdapter, a method that has gained prominence in the era of diffusion modeling. In fact, injection methods are not the focus of our research. We study how to construct and clean the style transfer dataset and how to design a unified framework to unify the style transfer framework, similar to how CSGO can realize image-driven and text-driven based style control tasks.
>
>
> ---
> **Q2：Concern of dataset**
>
> A: We agree with the reviewer that style transfer research should aim to enhance controllability, such as adjusting the degree of stylization or selectively transferring attributes like colors, strokes, or tonal styles. However, achieving more finer-grained style control is a much more challenging task, especially since end-to-end style transfer was already challenging prior to the collection of IMAGStyle. To achieve finer style control, we believe that it should be similar to the proposed ability of CSGO to learn all the style elements of a style image before it can further decouple and learn different types of style elements.
>
> The current implementation of CSGO can influence style transfer results by modulating the content injection strength scale, enabling a straightforward adjustment between more and less stylized outputs, as demonstrated in Figure 14.
>
> ---
>
> **Q3: Failure cases**
>
> A: Thanks to the reviewer’s suggestion, we have added the failure cases to the supplementary material. First, for real portrait stylization, as shown in the first row, there is a potential loss of facial identity. Portrait images can be difficult to collect due to the privacy issues involved, leading to some limitations in CSGO's style migration for real portraits.
> Second, despite incorporating styles into the ControlNet and base model, CSGO may still leak information, such as the original image's color. This phenomenon is due to the fact that the dataset still has insufficient pair data and needs to be further expanded using existing models (e.g. CSGO).
> In the future, we aim to enhance the CSGO framework in several ways. First, we plan to use CSGO in conjunction with LoRA to improve the portrait of the IMAGStyle dataset and enhance portrait stylization capabilities. Second, we will redesign and train the content encoder and style encoder to minimize content leakage and style leakage. However, we acknowledge that these improvements may not be achievable in the short term.
>
>
> ---
>
> **If our answers are more in line with your expectations, we kindly invite you to reconsider your initial rating, which is more confident for us to explore at a later stage.**

---

> ### Author Response · Authors · 2024-11-23
> **Follow-up**
>
> Dear Reviewer,
>
> We hope this message finds you well. We sincerely appreciate the time and effort you have dedicated to reviewing our submission. We have submitted our rebuttal and would like to follow up to inquire whether our responses have sufficiently addressed your concerns.
>
> Please let us know if you have any remaining questions or require additional clarification. We value your feedback and are eager to ensure our work meets the highest standards.
>
> Thank you again for your thoughtful insights and guidance.
>
> Best regards,
> CSGO Authors

---

> ### Author Response · Authors · 2024-11-28
>
> Dear Reviewer Vjbz:
>
> As today is the last day to revise the manuscript, I wanted to kindly follow up regarding the concerns you raised earlier. We have already provided detailed responses to address your feedback, but we have not yet received any further comments or suggestions.
>
> If there are any remaining points or clarifications needed, please feel free to let us know. We greatly value your insights and are eager to ensure the final manuscript meets your expectations.
>
> Thank you for your time and thoughtful consideration.
>
> Best regards,
>
> CSGO Authors

---

### Official Review · Reviewer_rbd7 · 2024-11-04

**Soundness:** 1
**Presentation:** 3
**Contribution:** 3
**Rating:** 5
**Confidence:** 5

**Summary:**

This paper was well written with clear structure and easy to understand. The paper firstly proposes a high quality and carefully cleaned dataset with 210k Content-StyleStylized Image Triplets.  Then, the paper proposes a new style transfer framework CSGO, which uses independent content and style feature injection modules to achieve high-quality image style transformations. Finally, a new score matrix named CAS was introduced to measure content loss after content-style transferred.

**Strengths:**

The open-source dataset of the article is valuable to the community. The experimental results reflected in the article are good.

**Weaknesses:**

The method proposed in the article is relatively simple and tends to be stacked. Although the author claims that this is an end-to-end approach, its innovation is insufficient.

**Questions:**

The feature injection amplification mentioned in the article are common methods, except for inject style features into Controlnet. What is the principle explanation for the operation mentioned in the paper —“The insight of this is to pre-adjust the style of the content image using style features making the output of the Controlnet model retain the content while containing the desired style features.” Actually, I can't see its importance from Fig 9. (2) W Content Control W/O style injection in ControlNet. More ablation study need to be supplemented with style similarity (CSD) and content alignment (CAS) Matrix.

---

> ### Author Response · Authors · 2024-11-21
> **Official Comment by Authors to Reviewer rbd7 (1/2)**
>
> We thank the reviewers for recognizing the **well-written**, **clear structure**, **easy to understand**, and **valuable dataset** of our article.
>
> ---
> **Q1：Clarification of the article's main contributions and model structure**
>
> A: The primary contribution of this study is the development of a construction pipeline and a cleaning method for style transfer datasets. Using this approach, we labeled a high-quality style transfer triad to enable effective end-to-end style transfer training. Given the challenges of collecting sufficient style transfer data from real-world scenarios, synthetic data was utilized to achieve this goal. In particular, as far as we know, the proposed IMAGStyle is the first large-scale style transfer dataset. To evaluate the effectiveness of the generated dataset, we implemented a straightforward yet highly effective framework, CSGO, which adheres to the mainstream paradigm. Despite this alignment, we carefully designed a content- and style-independent control framework. Consequently, CSGO unifies image-driven style transfer, text-driven stylized synthesis, and text-editor-driven stylized synthesis tasks.
>
> Finally, we would like to clarify that models are not assembled haphazardly. We have found that CSGO easily separates content and style branches, allowing for more precise content and style control to be implemented independently, such that CSGO can implement a wide range of text- and image-guided style control tasks. In addition, we believe that an easy-to-use CSGO framework is more likely to be extended and used by the community.
>
> ---
>
> **Q2: Explanation for operation mentioned in the paper —“The insight of this is to pre-adjust the style of the content image using style features making the output of the Controlnet model retain the content while containing the desired style features.**
>
> A: In CSGO, the Controlnet model is fixed-weighted to ensure that content information is as complete as possible after style transfer. Therefore, if the style features are only injected into the up block, **the original content features output by Tile controlnet will be directly injected into the up block and weaken the style information**. Therefore, we inject style features into the Controlnet model in advance so that the output of the controlnet contains pre-merged content and style features.
> In fact, the principle of injecting style features in controlnet is similar to that of the base stable diffusion model [1,2]. Fixing the weights of the base model can still adjust the style of the generated image, so it is also effective in the controlnet (i.e., the controlnet model, which structure is smliar with the base model).
> Finally, from the results of the supplemental quantitative experiments (see table below), the style similarity score CSD for style transfer was significantly improved after the injection of style features into controlnet.
>
> |       Metric       | (1) W/O Content Control | (2) W Content Control W/O style injection in ControlNet|     CSGO        |
> |:----------------:|:-----------:|:-----------:|:--------------------:|
> |     CSD     |0.5381|0.4873|  0.5146   |
> |  CAS   |1.7723|0.8372|          0.8386          |
> | Aesthetics Score |5.6325|5.5091| 5.5467 |
>
> [1]Hu Ye, Jun Zhang, Sibo Liu, Xiao Han, and Wei Yang. Ip-adapter: Text compatible image prompt adapter for text-to-image diffusion models. arXiv, 2023.
>
> [2] Haofan Wang, Qixun Wang, Xu Bai, Zekui Qin, and Anthony Chen. Instantstyle: Free lunch towards style-preserving in text-to-image generation. arXiv, 2024.
>
> ---

---

> > ### Author Response · Authors · 2024-11-21
> > **Official Comment by Authors to Reviewer rbd7 (2/2)**
> >
> > **Q3:  Ablation study results of  W Content Control W/O style injection?**
> >
> > A: Thanks to the reviewer for your valuable responses.
> > In response to the reviewers' suggestions, we incorporated quantitative results from the feature-injected ablation experiments. Additionally, we introduced the aesthetic score proposed by Reviewer 3 as a reference metric. The tabulated results reveal that the CSD score improves after stylistic features are injected into the ControlNet branch, indicating that the generated style aligns more closely with the input stylized image. Furthermore, the aesthetic score confirms that this modification does not diminish the visual appeal of the generated image.
> >
> >
> > |       Metric       | (1) W/O Content Control | (2) W Content Control W/O style injection in ControlNet|     CSGO        |
> > |:----------------:|:-----------:|:-----------:|:--------------------:|
> > |     CSD     |0.5381|0.4873|  0.5146   |
> > |  CAS   |1.7723|0.8372|          0.8386          |
> > | Aesthetics Score |5.6325|5.5091| 5.5467 |
> >
> >
> > From the quantitative results, it can be found that the lack of content control and the rise in CAS metrics indicate that textual prompts alone cannot maintain the original image content information. And after the introduction of content control, the CAS index decreases significantly, indicating that the content control branch plays the role of ensuring that the content is not lost. Meanwhile, it can be found that when the style features are injected into controlnet, the style features can be more significantly migrated to the content images, improving the quality of style transfer.
> >
> > ---
> >
> > **If our rebuttal better aligns with your expectations, we respectfully request that you reconsider your initial rating.**

---

> ### Author Response · Authors · 2024-11-23
> **Follow-up**
>
> Dear Reviewer,
>
> We hope this message finds you well. We sincerely appreciate the time and effort you have dedicated to reviewing our submission. We have submitted our rebuttal and would like to follow up to inquire whether our responses have sufficiently addressed your concerns.
>
> Please let us know if you have any remaining questions or require additional clarification. We value your feedback and are eager to ensure our work meets the highest standards.
>
> Thank you again for your thoughtful insights and guidance.
>
> Best regards,
> CSGO Authors

---

> ### Author Response · Authors · 2024-11-28
>
> Dear Reviewer rbd7
>
> As today is the last day to revise the manuscript, I wanted to kindly follow up regarding the concerns you raised earlier. We have already provided detailed responses to address your feedback, but we have not yet received any further comments or suggestions.
>
> If there are any remaining points or clarifications needed, please feel free to let us know. We greatly value your insights and are eager to ensure the final manuscript meets your expectations.
>
> Thank you for your time and thoughtful consideration.
>
> Best regards,
>
> CSGO Authors

---

### Author Response · Authors · 2024-11-22
**Reponse to All Reviewers**

We thank the reviewers for their feedback and comments. In particular, we are pleased that they found the article to be well motivated (zBGr), well structured (rbd7,Vjbz), and that the proposed IMAGStyle dataset is of high quality (y9qW), valuable (rbd7,Vjbz), and solid (Vjbz), and of extensive utility (qQFs, y9qW).  In addition, they perceived the style transfer results as good (rbd7) with high satisfaction (qQFs). Here we briefly outline the changes made to the manuscript and recurring points in the reviews.

**Changes to the Manuscript**

1) A discussion of failure cases was added in the supplemental material in Figure 2.

2) The presentation of CAS indicator filtering cases was added in Figure 1 of the 	Supplementary Material.

3) Added a comparison with the LoRA methodology, see Figure 3 in the Supplementary Material.

4) adjusted the arrangement of subfigure 2 of Fig. 1 to make it more reasonable.

5) Aligned the image alignment of Fig. 2, Fig. 5, Fig. 6, Fig. 7, Fig. 9, Fig. 10, and Fig. 11.

6) Aligned the text size in Figures 6, 7 and 8.

7) Fixed formatting issues with citep and citet.


**Q1：Clarification of the article's main contributions and model structure**


A: The primary contribution of this study is the development of a construction pipeline and a cleaning method for style transfer datasets. Using this approach, we labeled a high-quality style transfer triad to enable effective end-to-end style transfer training. Given the challenges of collecting sufficient style transfer data from real-world scenarios, synthetic data was utilized to achieve this goal. In particular, as far as we know, the proposed IMAGStyle is the first large-scale style transfer dataset. To evaluate the effectiveness of the generated dataset, we implemented a straightforward yet highly effective framework, CSGO, which adheres to the mainstream paradigm. Despite this alignment, we carefully designed a content- and style-independent control framework. Consequently, CSGO unifies image-driven style transfer, text-driven stylized synthesis, and text-editor-driven stylized synthesis tasks.

**Q2:Human evaluation results**

A: We added the human evaluation results. Setting: we randomly select 100 sets of results from the test set. Of these, 20 groups of portraits and 20 groups of sketches the rest were randomized. Subsequently, a user research experiment was conducted to compare CSGO with Styleshot-lineart, instantStyle, and Stylealigned respectively. Each group contains four generated results and the user selects the best result from the transfer quality.


|VS|       CSGO win       | Tile | CSGO loss |
|:----------------:|:----------------:|:-----------:|:-----------:|
|StyleShot|     58.5%     |21.4%|20.1%|
|Instanstyle|  64.2%   |20.6%|15.4%|
|StyleAligned| 67.0% |12.3%|10.7%|


**Q3：Difference with IP-Adapter, StyleAdapter, InstantID, InstantStyle.**


We show the differences between CSGO and the above methods in the table below. In particular, IP-Adapter and InstantID are different from the proposed tasks applicable to CSGO. Compared to StyleAdapter and InstantStyle, CSGO  support more diverse style control tasks, more detailed content and style control capabilities, and high-quality ternary style datasets.


|methods|	IP-Adapter[1]|	StyleAdapter[2]|	InstantID[3]|	InstantStyle[4]	|CSGO|
|:----------------:|:----------------:|:-----------:|:-----------:|:-----------:|:-----------:|
|Task	|Content Consistency maintenance|	Text-driven stylized synthesis|	ID Consistency Maintenance|	Text-driven stylized synthesis	|Image-driven style transfer, text-driven stylized synthesis, and text-editor-driven stylized synthesis tasks|
|Training data|	Reconstruction method, no pair data|	Reconstruction method, no pair data|	Reconstruction method, no pair data|	Reconstruction method, no pair	data|The proposed IMAGstyle, triplet data|
|Structural Properties	|Image features are injected all blocks by IP-Adapter| Image features by PCA are injected all blocks via IP-Adapter	| Face features are injected into all modules via IPA, identity net| Based on IPAdapter weights, image features are injected only into up_blocks.0.attentions.1	|Separates the content control and style control branches using IPadapter and controlnet. Style features are injected into controlnet and up_blocks through IPadapter respectively, and content features are controlled by controlnet and down_blocks|


[1]Ye, Hu, et al. "Ip-adapter: Text compatible image prompt adapter for text-to-image diffusion models." arXiv preprint arXiv:2308.06721 (2023).

[2]Wang, Zhouxia, et al. "Styleadapter: A single-pass lora-free model for stylized image generation." arXiv preprint arXiv:2309.01770 (2023).

[3]Wang, Qixun, et al. "Instantid: Zero-shot identity-preserving generation in seconds." arXiv preprint arXiv:2401.07519 (2024).

[4]Wang, Haofan, et al. "Instantstyle: Free lunch towards style-preserving in text-to-image generation." arXiv preprint arXiv:2404.02733 (2024).

---

### Author Response · Authors · 2024-11-27

We appreciate all reviewers valuable comments. We were wondering if our responses have addressed your concerns. Please let us know if you have additional questions. Thank you!

---

### Meta-Review · Area_Chair_npo9 · 2024-12-19

**Metareview:**

This work introduced a style encoder and an image encoder for style transfer task by using feature injection through selected layers, and collected a new 210K dataset for training such a model. While reviewers appreciate good results and the effort of dataset collection, there are also several major common concerns raised. It overall received three borderline reject and two borderline accept, while being a bit diverged but leaning towards negative. Two main weaknesses lie in the novelty (similarity with prior work like IP-adaptor or AdaIN) and the validity of collected (generated) dataset. The rebuttal unfortunately did not convince reviewers to obviously change their opinion, on the widely used idea of feature injection in prior arts and lack of evidence to prove the collected dataset is really useful. After checking all the comments and discussions, AC agrees that more contributions in method design and more in-depth analysis are needed to make this work more solid. Therefore a decision of reject is made and authors are encouraged to revise based on the comments for future resubmission.

**Additional Comments On Reviewer Discussion:**

Two main weaknesses lie in the novelty (similarity with prior work like IP-adaptor or AdaIN) and the validity of collected (generated) dataset. Authors do provide some explanations via rebuttal but AC agrees with reviewers that the difference of this work compared to other feature injection based method is incremental. In addition, three reviewers who gave the score of 5 (marginally below) are with highest confidence score 5. So AC puts more weight on their opinion compared to the other two reviewers who gave the score of 6. Thus AC made the reject decision.

---

### Decision · Program_Chairs · 2025-01-22

Reject